# JCGEL: Joint Color and Geometric Group Equivariant Convolutional Layer

## Abstract

Translation equivariance is one of the key factors for the widespread effectiveness of convolutional neural networks (CNNs) in computer vision. Building on this principle, group equivariant architectures have been extended beyond translations to encompass both color and geometric symmetries, which commonly arise in vision datasets. However, despite the commuting nature of their respective group actions, color and geometry have typically been addressed in isolation by theoretical and approximately equivariant approaches. In this paper, we introduce a *joint color and geometric group equivariant convolution layer (JCGEL)* via weight sharing across the commuting group actions. Our approach 1) improves robustness in imbalanced regimes, 2) yields factorized representations that separate color and geometric group-related factors, and 3) scales effectively to real-world datasets. To validate these effects, we instantiate the layer within standard CNNs and evaluate across long-tailed and biased datasets, disentanglement learning benchmarks, and real-world classification tasks, where our model consistently outperforms baselines. As a drop-in replacement for standard convolutional layers, JCGEL demonstrates generalization across a variety of vision tasks.

## 1 Introduction

Translation equivariance has been one of the primary factors enabling convolutional neural networks (CNNs) to extract spatial structure (LeCun et al., 1998; Kayhan & Gemert, 2020) and to achieve generalization across diverse computer vision tasks. To extend this benefit beyond translation, prior works have sustained interest in enforcing group equivariance, because many real-world variations are governed by symmetries. Formally, if an encoder $\psi$ is equivariant to a group $G$, then observing $x$ constrains $\psi(g \cdot x)$, even when $g \cdot x$ never appears in the data. In CNNs, translation equivariance implies that features learned for an object at one location transfer to the same object anywhere on the 2D plane LeCun et al. (1998); Kayhan & Gemert (2020). By the same principle, equivariance to other groups (rotations, scalings, and color transformations) yields consistent features for previously unseen variants. Group equivariant models have been shown to improve generalization across diverse areas include graph (Maron et al., 2018; Xu et al., 2024), robotics (Wu et al., 2023; Wang & Jörnsten, 2024; Qi et al., 2025), disentanglement learning (Higgins et al., 2018; Yang et al., 2021; Jung et al., 2024), self-supervised learning (Park et al., 2022; Yu et al., 2025), and equivariant layer modeling (MacDonald et al., 2021; Lengyel et al., 2023).

In the literature, approaches to propose group equivariant models have been proposed in two branches: 1) strict equivariant approaches that guarantee exact equivariance (Cohen & Welling, 2016a), and 2) soft approaches that encourage equivariance through less constrained kernel structures (Romero & Hoogendoorn, 2019) with training objectives (Kim et al., 2024). The first line of work, strict equivariant works, is theoretically equivariant to a specific group and has focused on geometric and color symmetries, which are pervasive in vision domains (Cohen & Welling, 2016a; Lengyel et al., 2023). Within the geometric line, early models target discrete groups (Cohen & Welling, 2016a) and have been extended to continuous geometric group, such as rotation, scaling, and Lie groups (Worrall & Welling, 2019; Qiao et al., 2023; Cohen & Welling, 2016b; Weiler et al., 2017; Sosnovik et al., 2019; MacDonald et al., 2021), with robustness in imbalanced environments. In parallel, color equivariant networks (Lengyel et al., 2023; Yang et al., 2025) address structured chromatic transformations and demonstrate strength under color imbalanced environments.

The second, recent works argue for the necessity of soft equivariant networks because real-world datasets rarely exhibit perfect symmetries (Wang et al., 2022; van der Ouderaa et al., 2022; Kim et al., 2024). On the geometric side, soft equivariant approaches relax exact constraints by regularizing canonical kernels with objectives, demonstrating advantages under asymmetric coverage (Wang et al., 2022; van der Ouderaa et al., 2022). In parallel, Kim et al. (2024) also validates that color soft equivariance via objective design, showing improved generalization on small and low-resolution real-world datasets. Taken together, strict and soft approaches underscore that geometric and color variations are ubiquitous and that their equivariant models are broadly useful. Nevertheless, to the best of our knowledge, no prior work offers a single convolutional operator that is jointly equivariant to commuting geometric (beyond translation, since standard CNNs already handle $T(2)$) and color groups under either strict or soft formulations.

To address this issue, we propose a joint color and geometric group equivariant layer (JCGEL). We first formalize the layer and prove equivariance to the direct product of group $G = (\mathbb{Z}^2 \rtimes G_{\text{geo}}) \times G_{\text{color}}$, where $\mathbb{Z}^2$ encodes planar translations, $G_{\text{geo}}$ acts on spatial coordinates (e.g., rotations/reflections), and $G_{\text{color}}$ acts in color space (e.g., hue shifts). We then introduce a $G$-equivariant batch normalization layer, enabling standard CNN architectures (e.g., ResNets (He et al., 2015b)). Finally, we validate from toy to real-world datasets and diverse vision tasks, showing consistent performance gains in imbalanced environment, disentanglement learning, and classification.

Our main contributions are as follows:

- **⚠ Bridging theory and practice.** We provide the first successful architectural realization of joint equivariance, solving non-trivial implementation challenges (e.g., channel interference) to translate the theoretical direct product into a working model.

- **Equivariant to both color and geometric groups.** We introduce a CNN architecture that is equivariant to the direct product group $G_{\text{geo}} \times G_{\text{color}}$, instantiated via a color and geometry equivariant convolutional layer.

- **Robustness under imbalance.** By sharing parameters across direct product group orbits (i.e., tying a canonical kernel via group actions), the model improves robustness in long-tailed and biased regimes.

- **Factorized representations.** The architecture yields a separable representation of color and geometry in latent space; we validate improved disentanglement through standard benchmarks and metrics.

- **Consistent gains on real-world datasets.** The approach scales to real-world datasets and delivers consistent performance on classification tasks.

## 2 RELATED WORKS

### 2.1 STRICT GROUP EQUIVARIANT CONVOLUTION LAYERS

Strict group equivariant CNNs generate all group-transformed filters from a canonical kernel or steerable-basis coefficients via the group action, enforcing weight tying and improving data efficiency and generalization. Geometry-focused approaches span discrete planar symmetries (Cohen & Welling, 2016a), continuous rotations (Worrall et al., 2016; Cohen & Welling, 2016b; Weiler et al., 2017), scaling (Sosnovik et al., 2019; Worrall & Welling, 2019), the Euclidean group $E(2)$ (Weiler & Cesa, 2019), and broader Lie groups (MacDonald et al., 2021; Qiao et al., 2023). Beyond geometric group, color group equivariant architectures have been proposed (Lengyel et al., 2023; Yang et al., 2025). However, to the best of our knowledge, a unified convolutional layer that achieves simultaneous equivariance to both geometric and color groups remains underexplored; most prior work enforces equivariance to either geometry or color, but not both jointly in a single layer.

### 2.2 SOFT GROUP EQUIVARIANT CONVOLUTIONAL LAYERS

Strict group equivariance assumes perfect symmetries in data, which is rarely met in practice. Soft equivariance approaches, therefore, relax architectural constraints and let the degree of equivariance be learned from data. In particular, statistical methods learn a distribution over group elements

Table 1: General group convolution definition of group equivariant CNNs. $G_o$, and $G_c$ denote geometric and color group.

| Model | Goup | Group Convolution Formula (Eq. 5) |
|---|---|---|
| Conv | $(\mathbb{Z}^2)$ | $\sum_{y \in \mathbb{Z}^2, \, c'} f_{c'}^\ell(y) \psi_{c'}^i(y-x)$ |
| $G_o$-CNNs | $\mathbb{Z}^2 \rtimes G_o$ | $\sum_{y \in \mathbb{Z}^2, \, c', \, h \in G_o} f_{c'}^\ell(y,h) \psi_{c'}^i(g^{-1}h(y-x))$ |
| $G_c$-CNNs | $\mathbb{Z}^2 \times G_c$ | $\sum_{y \in \mathbb{Z}^2, \, c', \, h \in G_c} f_{c'}^\ell(y,h) g^{-1}h \psi_{c'}^i(y-x)$ |
| Ours | $(\mathbb{Z}^2 \rtimes G_o) \times G_c$ | $\sum_{y \in \mathbb{Z}^2, \, c', \, h_o \in G_o, h_c \in G_c} f_{c'}^\ell(y,h_c,h_o) g_c^{-1}h_c \psi_{c'}^i(g_o^{-1}h_o(y-x))$ |

and sample group elements during the group convolution (Romero & Lohit, 2021), and other probabilistic/variational formulations further regularize or control the learned degree of equivariance via explicit objectives (Veefkind & Cesa, 2024; Kim et al., 2024). In addition, weighted mechanisms on the group fiber can emphasize a subset of symmetries (Romero & Hoogendoorn, 2019). More broadly, controlled departures from exact equivariance can be achieved through explicit regularization to accommodate imperfect symmetries. Despite these advances targeting asymmetric, real-world data, prior soft methods do not provide a single layer that is jointly equivariant to both color and geometric groups under a unified product-group action.

### 2.3 Non–Layer-Wise Approaches: Equivariant Inductive Bias via Objectives

Equivariance has also been encouraged by training objectives rather than by architecture, notably in self-supervised learning (SSL) and disentanglement learning. In SSL, recent methods inject transformation labels (Devillers & Lefort, 2022; Garrido et al., 2023) or enforce equal latent displacements for identically transformed pairs (Yu et al., 2025). In disentanglement, objectives are shaped so that latent coordinates align with subgroup actions, often via paired inputs in VAE frameworks (Jung et al., 2024; Yang et al., 2021; Keurti et al., 2022). These approaches inject equivariant bias through objectives and data pairing/composition, rather than by imposing per-layer group structure. Because our study targets layer-wise, drop-in convolutional operators under matched protocols, we do not include objective-level methods in head-to-head comparisons.

## 3 Preliminaries

In this section, we describe our notations, briefly introduce definitions of group action, equivariance, and group convolution.

**Group Action.** Let set $X$, and $(G, \circ)$ be a group, binary operation $\cdot : G \times X \to X$, then group action $\alpha : \alpha(g, x) = g \cdot x$ following properties:

- Identity: ② $\alpha(e, x) = x$, where $e \in G, \ x \in X$.

- Compatibility: $\forall g_1, g_2 \in G, \ x \in X, \ \alpha((g_1 \circ g_2), x) = \alpha(g_1, \alpha(g_2, x))$.

*(Dihedral Group Action)* The planar action uses the standard orthogonal representation $\rho(s, \theta) \in O(2)$ of the dihedral group $D_4 = \{(s, \theta) | s \in \{0, 1\}, \theta \in \mathbb{Z}_4\}$, where $\rho(s, \theta)$ is a rotation by $\theta \cdot \frac{\pi}{2}$ followed by a reflection group law:

$$(s, \theta) \cdot (s', \theta') = \left(s \oplus s', \theta + (-1)^s \theta' \pmod 4\right), \ (s, \theta)^{-1} = \left(s, -(-1)^s \theta \pmod 4\right), \quad (1)$$

where $\oplus$ is a modular arithmetic.

**Equivariant Map.** Given $X$ and $Y$ are $G$-set, and group action ③ $\alpha : G \times X \to X$, and $\rho : G \times Y \to Y$. Then a function $f : X \to Y$ is equivariant if

$$f(\alpha(g, x)) = \rho(g, f(x)). \quad (2)$$

④ **Standard Convolution.** In a standard CNN, feature map denoted $f^\ell : \mathbb{Z}^2 \to \mathbb{R}^{C^\ell}$ as a function that maps pixel locations $x$ to a $C^\ell$-dimensional vector. Then $f^\ell$ is convolved to filter

$\psi^\ell : \mathbb{Z}^2 \to \mathbb{R}^{C^\ell}$ as follows:

$$f^{\ell+1} = [f^\ell \star \psi^{\ell,i}](x) = \sum_{y \in \mathbb{Z}^2} \sum_{c=1}^{C^\ell} f_c^\ell(y) \psi_c^{\ell,i}(y - x), \tag{3}$$

⑤ where $\psi^{\ell,i}$ is a $i^{th}$ kernel of $\ell^{th}$ convolution layer. The standard CNNs is equivariant to the discrete translation group $(\mathbb{Z}^2, +)$. ⑥ The term $x - y$ in the convolution sum represents the discrete translation (shift) between the filter center $x$ and the input location $y$. The principle of equivariance inherent in this standard convolution can be extended to other transformation groups.

⑦ **Lifting Layer.** To generalize this concept, the group convolution is extended by replacing the discrete translation $x - y$ in the standard convolution operation with a general group action $g$. This specific layer is called the *lifting layer*, as it lifts the image features to the group domain:

$$f^{\ell+1} = [f^\ell \star \psi^{\ell,i}](g) = \sum_{y \in \mathbb{Z}^2} \sum_{c=1}^{C^\ell} f_c^\ell(y) \psi_c^{\ell,i}(g^{-1}y). \tag{4}$$

⑧ **Group Layer.** Then output feature map $f^\ell$ is a function on $G$ rather $\mathbb{Z}^2$, and is convolved with filter $\psi_c^{\ell,i}$ at $\ell^{th}$ layer, in what is referred to as *group layer*:

$$f^{\ell+1} = [f^\ell \star \psi^{\ell,i}](g) = \sum_{h \in G} \sum_{c=1}^{C^\ell} f_c^\ell(h) \psi_c^{\ell,i}(g^{-1}h). \tag{5}$$

# 4 METHOD: JOINT COLOR AND GEOMETRIC GROUP EQUIVARIANT CONVOLUTION LAYER

In this section, we prove that the proposed layer is group equivariant layer (i.e., satisfies Eq. 2) First, we formalize the lifting layer (Eq. 4) by specifying the associated group convolution and the group action invoked in Eq. 2. We then extend these definitions to the group layer (Eq. 5) and show that the layers preserve group equivariance.

Previous works have introduced color (hue shift) (Lengyel et al., 2023) or geometry ($D_4$) (Cohen & Welling, 2016a) equivariant layers separately. In contrast, we present a unified framework that composes these commuting symmetries within a single operator as summarized in Table 1. We define group $G = (\mathbb{Z}^2 \rtimes D_4) \times H_n$, $H_n \subset SO(3)$, the direct product of a geometric group $(\mathbb{Z}^2 \rtimes D_4)$ and a color group $H_n \subset SO(3)$. Here, $\mathbb{Z}^2$ denotes discrete translations on the image grid, $D_4$ the dihedral rotation–reflection group, and $H_n$ acts in color space; since the spatial and color actions operate on different domains, they commute.

## 4.1 LIFTING LAYER

**Joint Color and Geometric Group Convolution on Lifting Layer.** Given input image $f^\ell : \mathbb{Z}^2 \to \mathbb{R}^{C^\ell}$ and filters $\{\psi^{\ell,i}\}$, the lifting layer output $f^{\ell+1}(x, s, \theta, k)$ is obtained by a joint color and geometry convolution and indexed by spatial location $x$, color index $k$, and orientation $(s, \theta) \in D_4$ as follows:

$$[f^\ell \star \psi^{\ell,i}](x, s, \theta, k) = \sum_{y \in \mathbb{Z}^2} \sum_{c=1}^{C^\ell} \langle f_c^\ell(y), \ H_n(k) \, \psi_c^{\ell,i,(s,\theta)}(y - x) \rangle, \tag{6}$$

where $\psi^{i,(s,\theta)}(\zeta) := \psi^i\big(\rho(s,\theta)^{-1}\zeta\big)$ is the spatially transformed filter and $\langle \cdot, \cdot \rangle$ denotes the Euclidean inner product. Since $H_n(m)$ is orthogonal (Lengyel et al., 2023), for any $a, b \in \mathbb{R}^d$ we have $\langle H_n(m)a, \ b \rangle = \langle a, \ H_n(-m)b \rangle$.

**Group Action on Input Image Domain.** Then we introduce the operator $\mathcal{L}_g$ corresponding to the group action in Eq. 2. For $g = (t, s', \theta', m) \in G$ (translation $t \in \mathbb{Z}^2$, dihedral pose $(s', \theta')$, hue shift $m \in \mathbb{Z}_n$), we define the left action $\mathcal{L}_g^\ell$ on feature map of the $\ell^{th}$ layer as follows:

$$[\mathcal{L}_g^\ell f^\ell](x) = [\mathcal{L}_{(t,s',\theta',m)}^\ell f^0](x) = H_n(m) \, f^\ell\big(\rho(s',\theta')^{-1}(x - t)\big). \tag{7}$$

## 4.2 GROUP LAYER

**Joint Color and Geometric Group Convolution on Group Layer.** Similarly, a group-indexed feature $f^\ell : \mathbb{Z}^2 \times D_4 \times \mathbb{Z}_n \to \mathbb{R}^{C^\ell}$ is processed by group convolution with kernels $\{\psi^{\ell,i}\}$ defined on relative (group) indices, where $\ell > 0$. Introduced in Eq. 5, we then define the convolution on the group layer as follows:

$$[f^\ell \star \psi^{\ell,i}](x, s, \theta, k) = \sum_{y \in \mathbb{Z}^2} \sum_{s_1 \in \{0,1\}} \sum_{\theta_1 \in \mathbb{Z}_4} \sum_{m_1 \in \mathbb{Z}_n} \sum_{c=1}^{C^\ell} f_c^\ell(y, s_1, \theta_1, m_1)$$
$$\cdot \; \psi_c^{\ell,i}\Big(\rho(s,\theta)^{-1}(y-x), \; (s,\theta)^{-1}(s_1, \theta_1), \; (m_1 - k) \bmod n\Big). \tag{8}$$

⑨ Here, the hue shift $\mathbb{H}_n(m)$ difference between Eq. 6 is computed modulo $n$ rather than $H_n(k)$, which implements the cyclically permute for hue shift.

**Group Action on Group-Indexed Features.** For group-indexed feature map $f^\ell$, we define group action on group layer over $g = (t, s', \theta', m) \in G$ as follows:

$$[\mathcal{L}_g^\ell f^\ell](x, s, \theta, k) = f^\ell\big(\rho(s', \theta')^{-1}(x - t), \; (s', \theta')^{-1}(s, \theta), \; (k - m) \bmod n\big). \tag{9}$$

## 4.3 EQUIVARIANCE

The lifting and group layers of JCGEL is equivariant to group $G = (\mathbb{Z}^2 \rtimes D_4) \times H_n$, because these layers satisfy Eq. 2 as follows:

$$[\mathcal{L}_{(t,s',\theta',m)}^\ell f^\ell \star \psi^{\ell,i}](x, s, \theta, k) \tag{10}$$

$$= \sum_{z,c} \big\langle f_c^\ell(z), \; H_n(k-m) \, \psi^{\ell,i,(s \ominus s', (-1)^{s'}(\theta - \theta'))}\big(z - \rho(s', \theta')^{-1}(x - t)\big) \big\rangle \; (\because \text{Eq. } 6-7) \tag{11}$$

$$= [f^\ell \star \psi^i]\big(\rho(s', \theta')^{-1}(x - t), \; s \ominus s', \; (-1)^{s'}(\theta - \theta'), \; k - m\big) \; (\because \text{Eq. } 8) \tag{12}$$

$$= [\mathcal{L}_{(t,s',\theta',m)}^\ell [f^\ell \star \psi^{\ell,i}]](x, s, \theta, k) \; (\because \text{Eq. } 9), \tag{13}$$

where $\ominus$ is a modular arithmetic. Further details of proof for the direct product of groups are provided in Appendix B.1 and B.2.

## 4.4 IMPLEMENTATION

**Tensor Operations for Strict Equivariant.** We denote the filter $F^\ell$ instead of $\psi^\ell$ also feature $X^\ell$ rather than $f^\ell$ in Eq. 6 to represent the tensor shape. We store base spatial filters $F^\ell \in \mathbb{R}^{C^{\ell+1} \times C^\ell \times N^\ell \times H \times W}$, where $C^l$ is the number of base channels, $N^l = |H_n|$ (or euqal to $|\mathbb{H}_n|$) the number of color states, and $G^l = |D_n|$ the number of geometric states (quarter-rotations and flips). In the lifting layer for color equivariance, when $\ell = 0$, $N^\ell = 1$ then we extend kernel with hue-shfit matrix as introduced in Lengyel et al. (2023), then we get:

$$\tilde{F}_{c',n',:,1,u,v}^0 = H_n(k) F_{c',:,1,u,v}^0 \in \mathbb{R}^{C^{\ell+1} \times N^{\ell+1} \times C^\ell \times 1 \times H \times W}. \tag{14}$$

In the group layer, filter $\tilde{F}$ cyclically permuted copies of $F$ as follows:

$$\tilde{F}_{c',n',c,n,u,v}^\ell = F_{c',c,(n-n')\%k,u,v}^\ell \in \mathbb{R}^{C^{\ell+1} \times N^{\ell+1} \times C^\ell \times N^\ell \times H \times W}. \tag{15}$$

Then we implement JCGEL in the absolute rotation-and-flip manner for the geometric part rather than relative indexing of Eq. 8 because both methods are equivalent on the $D_4$ as shown in Cohen & Welling (2016a). Let $\mathcal{A}_{g'}$ denote the action of $g' \in D_n$ on spatial kernels $\tilde{F}^\ell$, $[\mathcal{A}_{g'}\tilde{F}^\ell](u) := \tilde{F}^\ell(\rho(g')^{-1}u)$ (rotate by $\theta' \cdot \frac{\pi}{2}$ and reflect if $s' = 1$). Then the group convolution is implemented as follows:

$$X_{c',n',g',:,:}^{l+1} = \sum_{c=1}^{C^l} \sum_{\Delta n \in H_n} \sum_{g \in D_n} \Big(\mathcal{A}_{g'}\tilde{F}_{c',n',c,\Delta n,1,:,:}^l\Big) \star X_{c,\Delta n,g,:,:}^l, \tag{16}$$

where $\star$ denotes 2D convolution. In the lifting layer, $\Delta n \in \{0, 1, \ldots, |H_n| - 1\}$ by the hue shift matrix (Eq. 6), and $\Delta n = n - n' \bmod k$ by the cyclic permutation operation in group layer (Eq. 8). This realizes Eq. 16 avoids explicit loops over $g$ and $g'$. For efficiency, we build the absolute kernel operator $\mathcal{A}_{g'}\tilde{F}^l$ for all $g' \in D_n$.

**Learnable Weight for Soft Equivariance**  ⑩ We employ a soft equivariant tensor operation to improve model generalization on datasets with imperfect symmetries van der Ouderaa et al. (2022); Romero & Hoogendoorn (2019). While strict equivariance enforces exact symmetry, real-world data often exhibits variations. Inspired by Romero & Hoogendoorn (2019), which relaxed equivariance by assigning weights to subgroups, we apply a learnable weighting mechanism to the geometric symmetry filters as follows:

$$X^{l+1}_{c',\,n',\,g',\,:,:} = \sum_{c=1}^{C^l} \sum_{n \in H_n} \sum_{g \in D_n} \tilde{w}_{g'} \Big( \mathcal{A}_{g'} \tilde{F}^l_{c',\,c,\,\Delta n,\,:,:} \Big) \star X^l_{c,\,n,\,g,\,:,:}, \tag{17}$$

where $\tilde{w}_{g'} = \frac{\text{softmax}(w_{g'}/\tau)}{\max(\text{softmax}(w_{g'}/\tau))}$, $w_{g'} \in \mathbb{R}^{|D_n|}$, and $\sum_{g'} w_{g'} = 1$.

**Group Equivariant Batch Normalization.**  When stacking JCGEL layers for large models, batch normalization is often necessary but it does not preserve equivariance. Motivated by Weiler & Cesa (2019), we normalize the group–indexed feature map $X^\ell \in \mathbb{R}^{B \times C \times |H_n| \times |D_n| \times H \times W}$. Further details are in the Appendix B.3.

## 5 EXPERIMENTS

First, we validate whether JCGEL is equivariant to both color (hue shift) and geometric ($D_4$) group in section 5.1, and robustness on an imbalanced environment in section 5.2. Then we investigate the effect of group-wise channel for factorized representations through disentanglement learning in section 5.3. Lastly, we evaluate our method in a classification task with real-world datasets to validate the impact in a practical environment in section 5.4. We focus on the impact of the equivariance of the direct product of groups rather than cutting-edge single-type group equivariant methods.

**Common Experimental Setting for Models.**  We replace standard CNN layers with group equivariant layers and ours: standard convolution (Conv) (LeCun et al., 1998), color equivariant convolution (CEConv) (Lengyel et al., 2023), $E(2)$-equivariant steerable CNN (E2CNN) Weiler & Cesa (2019), approximately equivariant networks (AE-Net) (Wang et al., 2022), Hue-4-Sat-3 Yang et al. (2025), and JCGEL. We set equivariant model parameters of $|G_{\text{geo}}| = |D_4|$ for E2CNN. Also $|G_{\text{geo}}| \in \{|D_4|, |D_2|, |C_4|\}$ with respect to imbalance, disentanglement, and classification tasks. $|H_n| = 3$ for CEConv and JCGEL, and $\tau \in \{1.0, 0.01\}$ with respect to imbalanced tasks and others for JCGEL. Also, we set $|G_{\text{geo}}| = |C_4|$, $L = 2$, and $\alpha \in \{0, 10^{-6}\}$ for AE-Net with relaxed group convolution.

### 5.1  ARE LIFTING AND GROUP LAYER OF JCGEL EQUIVARIANT TO GROUP $G$?

**Experimental Setting**  To validate equivariance to the hue shift and the dihedral group $D_4$, we generate 4,000 synthetic images.  ⑪ We set the synthetic images' size to $n \times n$ with $n \in \{17, 33, 65, 129\}$ to validate the robustness of image size, because the convolutional layer takes a diverse size of feature map during training and evaluation. We then evaluate equivariance using the mean-squared error: $Err = MSE([\mathcal{L}_g f \star \psi], [\mathcal{L}_g [f \star \psi]])$, where $f$ is a synthetic image, $g \in G_{\text{geo}} \times G_{\text{color}}$ with $g_{\text{geo}} \in D_4$ and $g_{\text{color}} \in H_n$. For each method, we evaluate both the lifting and group layers: we feed the synthetic images into the lifting layer, pass its output to the group layer, and compute the equivariance error as above. Further details are provided in Appendix C.1.

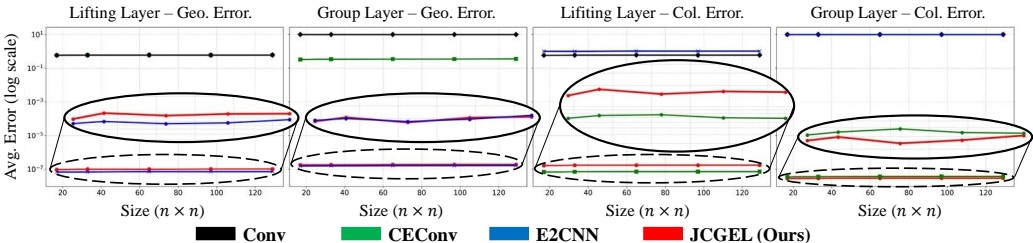

Figure 1: ⑫ Equivariant error evaluation. The x-axis shows the synthetic image side length ($H = W$), and the y-axis reports the equivariance error (lower is better).

**Equivariance Validation** As shown in Fig. 1, the lifting and group layers of JCGEL maintain equivariance to both the hue shift group $H_n$ and the dihedral group $D_4$. In particular, its geometric equivariance is on par with E2CNN, and its color equivariance remains competitive with CEConv, with variations on the order of $10^{-7}$ being negligible. Fig. 2a further shows that the output feature maps of JCGEL match at corresponding spatial locations across rotations (e.g., red/green boxes), whereas those of a standard convolutional anc CEConv layers vary substantially at the same object positions. Likewise, when applying a hue shift in feature space, the feature maps of two inputs related by the shift exhibit the expected cyclic correspondence across color-indexed channels, as shown in Fig. 2b.

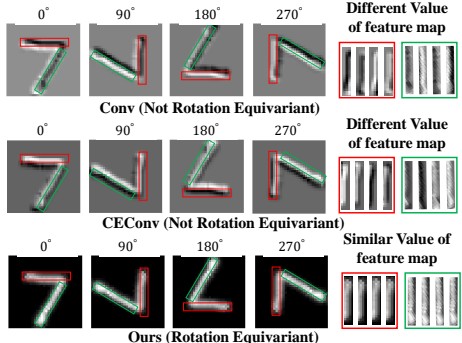
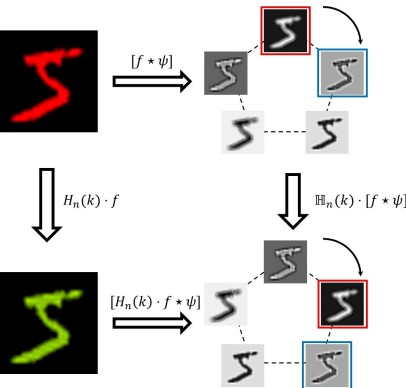

(a) Rotation equivariant test. The red and green boxes mark corresponding spatial locations before and after a $C_4$ rotation, matching feature-map values at these locations indicate rotation equivariance.

(b) Under a hue shift action, responses across color-indexed channels exhibit a cyclic shift, indicating equivariance to the color group.

Figure 2: ⑬ Color and $C_4$ group equivariant visualization with feature maps. JCGEL layer equivaraint to rotation and color simultaneously, whereas other baselines equivariant to a rotation or color.

## 5.2 COLOR AND ROTATION IMBALANCED ENVIRONMENT

Equivariance ties together all elements within a group orbit (a homogeneous space) of $G$: observing a few samples constrains the features of their symmetry-related counterparts $g \cdot x$ for all $g \in G$. Consequently, group equivariant models can generalize from limited evidence to unseen color/pose variants, a capability that is particularly valuable in imbalanced settings with scarce color or geometric coverage (Cohen & Welling, 2016a; Lengyel et al., 2023). Motivated by this, we evaluate robustness under controlled scarcity by constructing long-tailed and biased splits that deliberately reduce the availability of hue and rotation information.

**Experimental Setting for Imbalanced Environments** To validate the robustness of JCGEL in the absence of color and rotation information, we construct a long-tailed and biased rotated color MNIST (LeCun et al., 2012) dataset as follows:

- ⑭ To synthesize the Long-tailed Rotated-Color MNIST. Standard MNIST images are up-sampled to $64 \times 64$, embedded into a specific RGB channel $c \in \{0, 1, 2\}$, and rotated by discrete angles $\theta = 12k^\circ$. Crucially, we induce severe class imbalance by drawing the sample count $n_k$ for each (digit, color) pair from a power-law distribution:

$$n_k \sim \lceil \text{Power}(\alpha = 0.3) \cdot N_{\max} \rceil, \qquad (18)$$

while the test set remains balanced to fairly assess generalization. More details are in Appendix C.3 and Fig. 6.

- ⑮ We employ a hierarchical sampling scheme to synthesize the biased dataset (details in Appendix C.4). Crucially, the temperatures $\tau_c$ and $\tau_g$ govern the inter-class bias by determining the diversity of preferred color ($\mu_{c,y}$) and rotation ($\mu_{r,y}$) centers for each class $y$. Conditioned on these centers, sample counts are drawn via a multinomial distribution defined by the joint probability $P(c, r \mid y)$:

$$N_{c,r}^{(y)}{}_{c,r} \sim \text{Multinomial}\left(N_y, \text{vec}(P(c, r \mid y))\right). \qquad (19)$$

We evaluate seven-layer encoders and train with the Adam optimizer (Kingma & Ba, 2015) using an initial learning rate of $10^{-4}$ and a cosine-annealed schedule over 1,000 and 50 epochs with respect to the long-tailed and biased dataset. (warm up each epoch).

Table 2: Rotated Color MNIST (long-tailed, biased). Results averaged over three seeds. Red denotes the best score, and blue denotes the second-best. JCGEL* denotes the strict group equivariant network. Strong, moderate, and slight indicate bias level.

| Method | # param. ↓ | Long-Tailed ↑ | Biased ↑ | | | | |
|---|---|---|---|---|---|---|---|
| | | | $\tau_c, \tau_g = 0.5$ (strong) | $\tau_c, \tau_g = 5.0$ (moderate) | $\tau_c, \tau_g = 20.0$ (slight) | $\tau_c = 20.0, \tau_g = 10^{-9}$ (rotation biased) | $\tau_c = 10^{-9}, \tau_g = 20.0$ (color biased) |
| **Strict Equiv.** Conv. | 254.74K | 56.45(±0.26) | 36.47(±3.52) | 34.77(±2.06) | 29.67(±0.16) | 18.23(±0.38) | 21.32(±3.82) |
| CEConv. | 256.80K | 56.48(±1.60) | 45.50(±1.95) | 39.16(±2.89) | 29.85(±0.07) | 24.35(±2.18) | 31.99(±3.85) |
| E2CNN | 250.81K | 50.88(±2.05) | 41.17(±10.07) | 35.82(±6.69) | 28.22(±0.20) | 18.08(±0.31) | 17.96(±0.22) |
| Hue-4-Sat-3 | 322.59K | 52.17(±1.08) | 45.38 (±1.85) | 38.40(±1.76) | 35.08(±1.50) | 19.42(±1.00) | 20.46(±1.24) |
| JCGEL* | **184.82K** | **59.26**(±0.14) | **75.49**(±0.49) | **74.88**(±0.88) | **75.14**(±1.32) | **68.60**(±0.93) | **68.74**(±0.97) |
| **Soft Equiv.** AE-Net | 223.39K | 57.64(±0.74) | 42.82(±7.96) | 44.13(±5.18) | 37.73(±1.32) | 22.88(±1.43) | 25.38(±9.76) |
| JCGEL | **184.82K** | **57.87**(±0.78) | **75.43**(±0.86) | **75.69**(±0.51) | **75.49**(±0.65) | **74.64**(±0.64) | **74.10**(±1.31) |

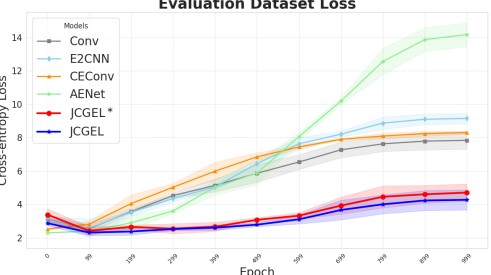

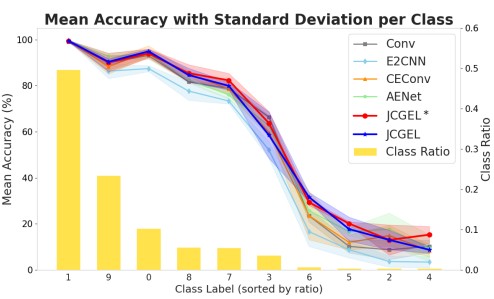

(a) Evaluation cross-entropy loss during training.  (b) Accuracy and portion per class.

Figure 3: Visualization of Long-tailed rotated color MNIST Results.

**Results under Imbalance** Consistent with our objective, JCGEL generalizes from scarce evidence to unseen hue and rotation variants as shown in Table 2. Across all bias levels (Strong/Moderate/Slight), JCGEL and its strict variant, JCGEL* outperform baselines. In the extreme color and rotation bias setting ($\tau_{c,g} = 0.5$, with training dominated by red), JCGEL correctly predicts blue/green instances at test time even though those hues are essentially unobserved during training. On long-tailed splits, gains concentrate on tail classes, and JCGEL shows the smallest increase in test loss during training, indicating improved generalization to long-tailed classes as shown in Fig. 3. We also observe a bias-dependent preference: under strong skew, the strict model JCGEL* surpasses soft approaches, whereas under slight skew the soft variant outperforms the strict model, as shown in Table 2. The same result appears for AE-Net (soft) and CEConv (strict). Overall, these results support that joint color and geometric equivariance is most beneficial in imbalanced regimes with scarce hue and rotation coverage, lifting tail-class accuracy while maintaining robust generalization.

## 5.3 DISENTANGLEMENT LEARNING

Following the group-theoretic view, a representation is disentangled when latent coordinates factorize along subgroup actions, so that each block contains only its associated latent factors of variation (Higgins et al., 2018). Motivated by this definition, we test whether the group-wise channel structure of group equivariant models (including ours) promotes such factorization.

**Experimental setting of Disentanglement Learning** We evaluate disentanglement on 3D Shapes (Burgess & Kim, 2018) and MPI3D (Eslami et al., 2018). For each method, we replace the VAE encoder's four convolutional layers with group-equivariant counterparts (CEConv, E2CNN, AE-Net, and JCGEL) and train using Adam (learning rate $8 \times 10^{-4}$), a batch size of 512, and 500,000 training iterations. We report standard metrics—BetaVAE score (Higgins et al., 2017), FVM (Kim & Mnih, 2018), MIG (Chen et al., 2018), SAP (Kumar et al., 2018), and DCI (Eastwood & Williams, 2018). Additional architectural and training details are provided in Appendix C.5.

Table 3: Disentanglement performance on 3D Shapes and MPI3D datasets. Results are reported as mean ± std over three seeds. Bold text indicates scores higher than all baseline models.

| Method | # param. | 3D Shapes | | | | | |
| --- | --- | --- | --- | --- | --- | --- | --- |
| | | beta-VAE ↑ | FVM ↑ | MIG ↑ | SAP ↑ | DCI-Dis. ↑ | DCI-Com. ↑ |
| **Strict Equiv.** | | | | | | | |
| Conv. | 1.51M | 77.33(±7.57) | 71.46(±4.38) | 31.79(±6.18) | 6.57(±2.48) | 46.50(±3.95) | 47.53(±4.43) |
| CEConv. | 1.78M | **92.67**(±3.06) | 83.88(±1.44) | 44.74(±8.16) | 7.22(±2.35) | 59.66(±4.44) | 61.44(±4.18) |
| E2CNN | 1.60M | 89.33(±10.07) | 82.13(±6.71) | 43.53(±10.02) | **9.15**(±1.51) | 52.44(±8.71) | 53.78(±8.78) |
| Hue-4-Sat-3 | 2.57M | 80.00(±9.17) | 79.88(±2.25) | 28.59(±6.89) | 5.90(±1.51) | 45.94(±4.61) | 47.66(±4.97) |
| JCGEL* | 1.52M | **95.33**(±6.43) | **83.96**(±8.49) | **44.61**(±15.66) | **8.90**(±2.71) | **59.94**(±12.07) | **64.11**(±7.16) |
| **Soft Equiv.** | | | | | | | |
| AE-Net | 1.62M | 79.00(±1.41) | 52.38(±2.30) | 7.25(±5.08) | 2.00(±1.03) | 25.49(±7.50) | 25.56(±7.49) |
| JCGEL | 1.52M | **92.67**(±7.02) | **87.67**(±4.57) | **56.72**(±3.94) | 8.55(±1.90) | **66.86**(±4.74) | **67.82**(±4.94) |

| Method | # param. | MPI3D | | | | | |
| --- | --- | --- | --- | --- | --- | --- | --- |
| | | beta-VAE ↑ | FVM ↑ | MIG ↑ | SAP ↑ | DCI-Dis. ↑ | DCI-Com. ↑ |
| **Strict Equiv.** | | | | | | | |
| Conv. | 1.51M | 48.67(±9.45) | 39.50(±4.75) | 3.85(±0.51) | 2.57(±0.88) | 18.98(±1.89) | 27.68(±1.08) |
| CEConv. | 1.78M | 58.00(±7.21) | 39.58(±8.49) | 3.79(±1.08) | 2.09(±0.78) | 18.79(±2.99) | 27.27(±1.64) |
| E2CNN | 1.60M | 49.00(±18.38) | 41.44(±5.21) | 3.62(±1.55) | 1.37(±0.86) | 21.60(±1.18) | 27.54±1.91) |
| Hue-4-Sat-3 | 2.57M | 51.33(±1.15) | 42.42(±6.25) | 5.05(±1.52P) | 3.37(±0.74) | 20.77(±0.86) | 28.65(±0.90) |
| JCGEL* | 1.52M | **69.33**(±1.15) | **46.79**(±3.50) | **13.80**(±2.65) | **7.86**(±2.00) | **31.85**(±3.05) | **31.51**(±1.91) |
| **Soft Equiv.** | | | | | | | |
| AE-Net | 1.62M | 49.00(±18.38) | 41.44(±5.21) | 3.63(±1.55) | 1.37(±0.86) | 21.60(±1.18) | 27.54(±1.91) |
| JCGEL | 1.52M | **60.67**(±2.31) | **45.75**(±3.56) | **12.27**(±12.05) | **6.20**(±5.89) | **23.27**(±4.04) | **31.93**(±5.56) |

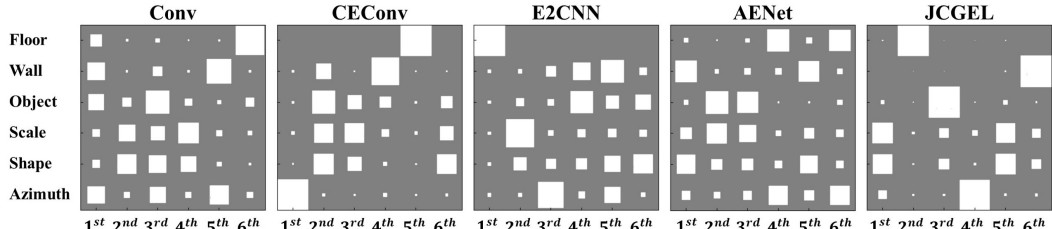

Figure 4: DCI matrix visualization: The DCI matrix shows the feature importance $r_{k,j}$, how strongly the latent vector $z_j$ predicts the ground-truth factor $v_k$, where $z_j \in \{1, 2, \ldots, 6\}$ and $v_k \in$ {Floor, Wall, Object, Scale, Shape, Azimuth} with 3D Shapes. The better disentangled representation appears as a sparse matrix with a few large, isolated cells.

**Results of Disentanglement Learning** Across both 3D Shapes and MPI3D, our method outperforms all baselines in terms of disentanglement scores as shown in Table 3. Notably, 3D Shapes contains richer color variation, while MPI3D emphasizes geometric variation. Despite these differing factor profiles, our model yields robust gains on both datasets. In contrast, E2CNN tends to benefit primarily when geometric variation dominates, and CEConv when color variation dominates, indicating a dependency on dataset composition. As shown in Fig. 4, visualizations further show that our latent coordinates align sparsely with individual factors, supporting the intended effect of the group-wise channel design. Finally, although recent work introduces objectives to learn equivariance, we find that simply replacing encoder layers with our equivariant counterparts already delivers consistent improvements in disentanglement quality.

## 5.4 CLASSIFICATION IN REAL-WORLD DATASETS

While many equivariant layers are designed as drop-in replacements for standard convolutions, evidence on large-scale, real-world settings remains: existing evaluations often focus on small or low-resolution datasets (Kim et al., 2024), and the reported gains can be sensitive to model configurations (Lengyel et al., 2023; Yang et al., 2025). To identify which approaches truly scale beyond controlled benchmarks, we run a comparative classification study on real-world datasets, evaluating group equivariant models and ours.

Table 4: Classification accuracy on real-world datasets.

| original dataset | Layer | # params. | EuroSAT (5.6K) | CIFAR100 (60K) | Pets (8.2K) | Flowers (9.1K) | Aircraft (13K) | STL10 (15.6K) | Food101 (101K) | ImageNet (1.2M) |
| --- | --- | --- | --- | --- | --- | --- | --- | --- | --- | --- |
| **Strict Equiv.** | Conv. | 43.59M | 97.46(±0.34) | 76.20(±0.24) | 74.86(±1.28) | 52.99(±1.23) | 53.02(±0.24) | 85.24(±0.33) | 81.26(±0.26) | 64.77 |
| | CEConv. | 42.02M | 97.75(±0.14) | 76.10(±0.14) | 68.76(±0.54) | 54.01(±1.56) | 52.60(±0.81) | 84.40(±1.38) | 81.45(±0.31) | 67.83 |
| | E2CNN | 36.88M | 95.38(±0.32) | 77.29(±0.01) | 67.41(±0.86) | 55.62(±1.36) | 50.26(±5.88) | 85.30(±0.09) | 79.79(±0.26) | 64.73 |
| | Hue-4-Sat-3 | 37.26M | 97.51(±0.10) | | 65.39(±3.42) | 54.16(±0.61) | 52.95(±2.66) | 79.23(±0.19) | 79.38(±0.21) | 64.45 |
| | JCGEL* | 41.03M | 97.69(±97.69) | 77.33(±0.19) | 75.25(±0.61) | 54.61(±0.10) | **54.61**(±0.99) | 85.49(±0.35) | 82.60(±0.16) | 69.52 |
| **Soft Equiv.** | AE-Net | 46.28M | **97.83**(±0.15) | 72.99(±0.43) | 66.00(±0.36) | 48.63(±1.91) | 48.67(±1.13) | 82.43(±1.20) | 82.04(±0.11) | 69.54 |
| | JCGEL | 41.03M | 97.70(±0.18) | **77.51**(±0.45) | **76.08**(±0.80) | **56.73**(±1.37) | 54.11(±0.92) | **85.54**(±0.26) | **82.62**(±0.38) | 70.43 |

Table 5: ⑰ Classification accuracy on real-world datasets (augmeneted dataset).

| Aug. dataset | Layer | EuroSAT (5.6K) | CIFAR100 (60K) | Pets (8.2K) | Flowers (9.1K) | Aircraft (13K) | STL10 (15.6K) | Food101 (101K) |
|---|---|---|---|---|---|---|---|---|
| Strict Equiv. | Conv. | 53.15(±1.18) | 26.73(±0.53) | 32.33(±0.36) | 10.69(±0.20) | 12.93(±1.33) | 46.95(±0.83) | 21.58(±0.19) |
| | CEConv. | 49.87(±2.20) | 24.60(±0.41) | 32.25(±1.40) | 11.88(±0.30) | 12.76(±0.53) | 46.80(±1.06) | 19.57(±0.74) |
| | E2CNN | 59.26(±0.39) | 24.21(±1.59) | 23.56(±0.50) | 11.96(±0.71) | 13.1(±1.12) | 44.90(±1.27) | - |
| Soft Equiv. | AE-Net | 55.85(±0.92) | 24.89(±0.28) | 32.57(±0.01) | 10.79(±0.02) | 11.65(±0.24) | 46.51(±1.18) | 22.66(±1.36) |
| | JCGEL | **66.48**(±1.16) | **51.35**(±0.56) | **58.31**(±1.23) | **22.72**(±0.88) | **13.56**(±1.87) | **46.97**(±0.72) | **23.52**(±0.71) |

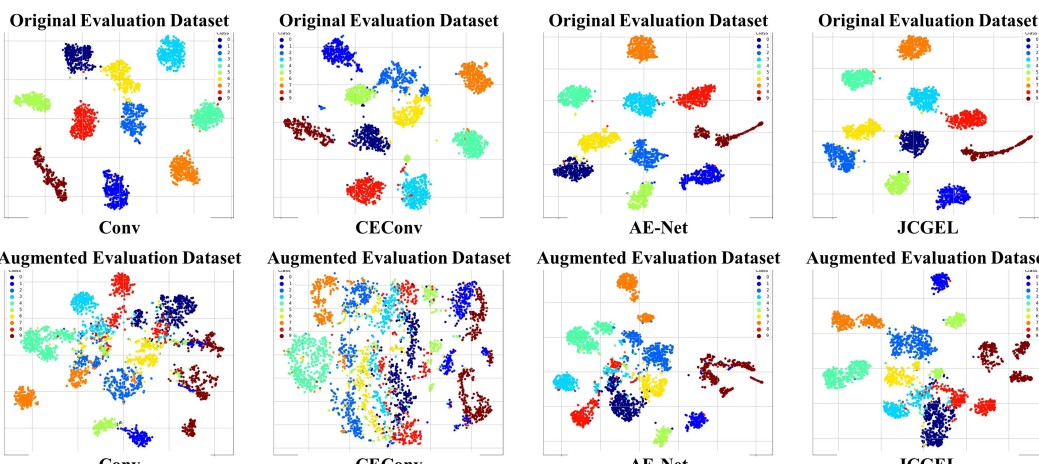

Figure 5: EuroSAT feature-map visualization on original and augmented test images. The augmented set applies a random composite transformation at evaluation time: a continuous hue shift over the full hue circle and an in-plane rotation with angle $\theta \sim \mathcal{U}[-\pi, \pi]$.

**Experimental Setting of Real-World Classification**   We report top-1 accuracy on real-world datasets (Helber et al., 2019; Krizhevsky & Hinton, 2009; Parkhi et al., 2012; Nilsback & Zisserman, 2008; Maji et al., 2013; Coates et al., 2011; Bossard et al., 2014). For each method, we replace the convolutional layers of a ResNet-18 (He et al., 2015b) with the candidate group equivariant operator and adjust block widths to keep parameter counts comparable across models. Further architectural and training details are provided in Appendix C.6.

**Results of Accuracy and Robustness to Hue Shift and Rotation Variation**   Across the seven real-world datasets, JCGEL delivers consistent accuracy gains over the vanilla convolutional baseline and other group equivariant layers, with the exception of EuroSAT, as shown in Table 5. By contrast, alternative group equivariant models (E2CNN, CEConv, and AE-Net) exhibit dataset-dependent behavior, sometimes improving over standard convolutions but often falling short. Under composite, continuous hue shifts and in-plane rotations, the augmented dataset yields severely disrupted t-SNE embeddings for Conv, CEConv, and AE-Net—class boundaries blur relative to the original set as shown in Fig. 5. In contrast, JCGEL shows a distributional shift yet maintains clear inter-class separation. Taken together, these findings align with our objective: replacing the layer that enforces joint color and geometry equivariance provides the most reliable inductive bias among the evaluated methods for real-world classification.

## 6   CONCLUSION

In this paper, we address the lack of a drop-in convolutional operator that achieves simultaneous equivariance to commuting geometric (beyond translation) and color transformations, a capability needed for the computer vision domain. We propose JCGEL, a joint color and geometric group equivariant convolutional layer that can replace standard convolutions in common backbones. With only this substitution, we observe improvements on imbalanced environments, disentanglement learning, and real-world classification. These results suggest that enforcing equivariance to a direct product of groups is better suited to real-world image grids than targeting a single continuous group and has the potential to address a broader range of tasks.

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

# A   ROLE OF THE LLM

Throughout this study, we employed an LLM at the sentence level to assess grammar, strengthen within-paragraph cohesion, and ensure that our intended content was clearly conveyed.

# B   METHOD DETAILS

**Notation differences from the main text.** In the proofs we work in the Euclidean group $E(2) = \mathbb{R}^2 \rtimes O(2)$ and then specialize to the dihedral subgroup $D_n$. This choice is purely notational: $D_n \leq E(2)$, and the parameterization we use (translations and planar rotations/reflections) is the same in both settings, so establishing equivariance for $E(2)$ yields the $D_n$ case as a direct corollary.

To simplify expressions, we drop layer superscripts and other adornments on feature maps, filters, and group actions. In the lifting layer we write the image domain feature map and filter as $f : \mathbb{Z}^2 \to \mathbb{R}^{C^\ell}$ and $\psi : \mathbb{Z}^2 \to \mathbb{R}^{C^\ell}$, with input-domain action $\alpha$. In the group layer we use capital letters $F : G \to \mathbb{R}^{C^\ell}$ and $\Psi : G \to \mathbb{R}^{C^\ell}$, and denote the induced feature-space action by $\rho$. When the domain is clear, we further omit subscripts on convolution/cross-correlation operators for readability.

## B.1   LIFTING LAYER

**Setup and notation.** Let $f : \mathbb{Z}^2 \to \mathbb{R}^{C^\ell}$ be an input feature map with $C^\ell$ channels. Let $\psi^i : \mathbb{Z}^2 \to \mathbb{R}^{C^\ell}$ be a learnable filter for $i \in \{1, \ldots, |C^{\ell+1}|\}$. Denote by $H_n(k) \in \mathbb{R}^{d_c \times d_c}$ an orthogonal hue-rotation matrix (Lengyel et al., 2023) for $k \in \mathbb{Z}_n$. For geometry, write group elements of $O(2)$ as $(s, \theta)$ with $s \in \{0, 1\}$ (flip bit) and $\theta \in \mathbb{R}/2\pi\mathbb{Z}$ (a rotation angle). Let $R(\theta) \in SO(2)$ be the counter-clockwise rotation by angle $\theta$, and let $F$ be a fixed reflection (e.g., $F = \mathrm{diag}(1, -1)$). We use the faithful $2 \times 2$ orthogonal representation

$$\rho(s, \theta) = \begin{cases} R(\theta), & s = 0, \\ R(\theta)\, F, & s = 1. \end{cases} \tag{20}$$

The $O(2)$ group law is

$$(s_1, \theta_1) \cdot (s_2, \theta_2) = \big(s_1 \oplus s_2,\ \theta_1 + (-1)^{s_1}\theta_2 \mod 2\pi\big), \tag{21}$$

where $\oplus$ is addition modulo 2, and inverses are

$$(s, \theta)^{-1} = \big(s,\ -(-1)^s\theta \mod 2\pi\big). \tag{22}$$

We target the direct-product group

$$G = E(2) \times H_n = (\mathbb{Z}^2 \rtimes O(2)) \times H_n, \tag{23}$$

where $(s, \theta) \in O(2)$ acts on translations by $\rho(s, \theta)\, t$.

**Group Action on Inputs.** For $g = (t, s', \theta', m) \in G$ (translation $t \in \mathbb{Z}^2$, flip $s' \in \{0, 1\}$, rotation $\theta' \in \mathbb{R}/2\pi\mathbb{Z}$, hue shift $m \in \mathbb{Z}_n$), define the left action

$$[\mathcal{L}_g f](x) = [\mathcal{L}_{(t,s',\theta',m)} f](x) = H_n(m)\, f\big(\rho(s', \theta')^{-1}(x - t)\big). \tag{24}$$

Since $H_n(m)$ is orthogonal (Lengyel et al., 2023), for any $a, b \in \mathbb{R}^{d_c}$ we have $\langle H_n(m)a,\, b \rangle = \langle a,\, H_n(-m)b \rangle$.

**Induced Output Action.** Let $F(x, s, \theta, k)$ be an output feature. The induced left action on outputs is

$$[\mathcal{L}^c_{(t,s',\theta',m)} F](x, s, \theta, k) := F\Big(\rho(s', \theta')^{-1}(x - t), \ s \ominus s', \ \mathrm{wrap}\big((-1)^{s'}(\theta - \theta')\big), \ k - m\Big), \quad (25)$$

where $\ominus$ is subtraction in $\mathbb{Z}_2$ (which equals $\oplus$) and $\mathrm{wrap}(\cdot)$ maps angles to $[0, 2\pi)$ (any fixed $2\pi$-periodic choice suffices).

**Proof: Details of $E(2) \times H_n(k)$ Equivariance on lifting layer.** We show $[\mathcal{L}_g f \star \psi^i] = \mathcal{L}^c_g [f \star \psi^i]$ for all $g = (t, s', \theta', m) \in G$. By definition and orthogonality of $H_n$,

$$[\mathcal{L}_{(t,s',\theta',m)} f \star \psi^i](x, s, \theta, k) = \sum_{y \in \mathbb{Z}^2} \sum_{c=1}^{C^\ell} \big\langle [L_g f_c](y), H_n(k) \psi^{i,(r,\theta)}_c(y - x) \big\rangle \quad (26)$$

$$= \sum_{y \in \mathbb{Z}^2} \sum_{c=1}^{C^\ell} \big\langle H_n(m) f_c(\rho(s', \theta')^{-1}(y - t)), H_n(k) \psi^{i,(s,\theta)}_c(y - x) \big\rangle \quad (27)$$

$$= \sum_{y,c} \big\langle f_c(\rho(s', \theta')^{-1}(y - t)), H_n(k - m) \psi^{i,(s,\theta)}_c(y - x) \big\rangle \quad (28)$$

$$= \sum_{y,c} \big\langle f_c(\rho(s', \theta')^{-1}(y - t)), H_n(k - m) \psi^i_c\big(\rho(s, \theta)^{-1}(y - x)\big) \big\rangle \quad (29)$$

Let $z = \rho(s', \theta')^{-1}(y - t)$ so $y = \rho(s', \theta') z + t$. Then,

$$[\mathcal{L}_{(t,s',\theta',m)} f \star \psi^i](x, s, \theta, k) = \sum_{z,c} \big\langle f_c(z), H_n(k - m) \psi^i\big(\rho(s, \theta)^{-1}(\rho(s', \theta') z + t - x)\big) \big\rangle$$

$$= \sum_{z,c} \Big\langle f_c(z), H_n(k - m) \psi^i\Big(\rho(q_{\mathrm{rel}}) \big[z - \rho(s', \theta')^{-1}(x - t)\big]\Big) \Big\rangle_c, \quad (30)$$

where we used the $O(2)$ group property to factor the argument via the relative pose

$$q_{\mathrm{rel}} := (s, \theta)^{-1} \cdot (s', \theta') \ = \ \big(s \oplus s', \ -(-1)^s \theta + (-1)^s \theta'\big) \ = \ \big(s \oplus s', \ (-1)^s(\theta' - \theta)\big). \quad (31)$$

Equivalently, we write $\psi^i\Big(\rho(q_{\mathrm{rel}})[\cdot]\Big) = \psi^{i,(q_{\mathrm{rel}}^{-1})}(\cdot)$ so that

$$[\mathcal{L}_{(t,s',\theta',m)} f \star \psi^i](x, s, \theta, k) = \sum_{z,c} \Big\langle f_c(z), H_n(k - m) \psi^{i,(q_{\mathrm{rel}}^{-1})}\big(z - \rho(s', \theta')^{-1}(x - t)\big) \Big\rangle_c. \quad (32)$$

Now, unpack $q_{\mathrm{rel}}^{-1}$ using equation 22:

$$q_{\mathrm{rel}} = \big(s \oplus s', \ (-1)^s(\theta' - \theta)\big) \quad \Rightarrow \quad q_{\mathrm{rel}}^{-1} = \Big(s \oplus s', \ -(-1)^{s \oplus s'}(-1)^s(\theta' - \theta)\Big) = \Big(s \oplus s', \ (-1)^{s'}(\theta - \theta')\Big),$$

where angles are understood modulo $2\pi$. Hence

$$[\mathcal{L}_{(t,s',\theta',m)} f \star \psi^i](x, s, \theta, k) \quad (33)$$

$$= \sum_{z,c} \big\langle f_c(z), H_n(k - m) \psi^{i,(s \ominus s', (-1)^{s'}(\theta - \theta'))}\big(z - \rho(s', \theta')^{-1}(x - t)\big) \big\rangle \quad (34)$$

$$= [f \star \psi^i]\big(\rho(s', \theta')^{-1}(x - t), \ s \ominus s', \ \mathrm{wrap}((-1)^{s'}(\theta - \theta')), \ k - m\big) \ (\because \text{Eq. 6}) \quad (35)$$

$$= [\mathcal{L}^c_{(t,s',\theta',m)} [f \star \psi^i]](x, s, \theta, k) \ (\because \text{Eq. 25}), \quad (36)$$

which proves $E(2)$-equivariance jointly with hue shift.

## B.2   Color and $O(2)$ Group Layer

**Group structure.**   The orthogonal group $O(2)$ can be expressed as the semidirect product $SO(2) \rtimes \mathbb{Z}_2$. Each element is written $(s, \theta)$ with $s \in \{0, 1\}$ (flip) and $\theta \in S^1 = \mathbb{R}/2\pi\mathbb{Z}$ (rotation angle). Its law and inverse are

$$(s_1, \theta_1) \cdot (s_2, \theta_2) = \big(s_1 \oplus s_2, \ \theta_1 + (-1)^{s_1}\theta_2 \bmod 2\pi\big), \tag{37}$$

$$(s, \theta)^{-1} = \big(s, \ -(-1)^s\theta \bmod 2\pi\big). \tag{38}$$

The color group $H_n = \mathbb{Z}_n$ acts via an orthogonal representation $H_n(k)$, $k \in \{0, \ldots, n-1\}$, with cyclic composition $k_1 \oplus k_2 = (k_1 + k_2) \bmod n$. Hence, the total group is

$$G = (\mathbb{Z}^2 \rtimes O(2)) \times H_n.$$

**Feature domains.**   A group-layer feature map is

$$F: \ \mathbb{Z}^2 \times \{0, 1\} \times S^1 \times \mathbb{Z}_n \ \longrightarrow \ \mathbb{R}^{C^\ell}.$$

That is, each feature is indexed by spatial location $x \in \mathbb{Z}^2$, flip $s$, rotation $\theta$, and hue index $k$. A learnable filter $\psi^i$ (for output channel $i$) is defined on relative indices

$$\Psi^i: \ \mathbb{Z}^2 \times \{0, 1\} \times S^1 \times \mathbb{Z}_n \ \longrightarrow \ \mathbb{R}^{C^\ell}.$$

**Group correlation.**   We follow the group correlation (Cohen & Welling, 2016a) as introduced $[F \star \Psi](g) = \sum_{h \in G} f(g)\Psi(g^{-1}h)$. The group correlation producing the output at $(x, s, \theta, k)$ is

$$\begin{aligned}
[F \star \Psi^i](x, s, \theta, k) = \sum_{y \in \mathbb{Z}^2} \sum_{s_1 \in \{0,1\}} \int_0^{2\pi} \sum_{m_1 \in \mathbb{Z}_n} \sum_{c=1}^{C^\ell} & F_c(y, s_1, \theta_1, m_1) \\
& \cdot \ \Psi_c^i\Big(\rho(s, \theta)^{-1}(y - x), \ (s, \theta)^{-1}(s_1, \theta_1), \ (m_1 - k) \bmod n\Big) \frac{d\theta_1}{2\pi}.
\end{aligned} \tag{39}$$

Here, the hue difference is computed modulo $n$, which implements the rolling structure of hue shift. In practice, the continuous integral $\int_0^{2\pi}$ is approximated by a uniform sample sum $\frac{1}{Q}\sum_{\theta_1}$ with $Q$ orientations.

**Group action on inputs.**   For $g = (t, s', \theta', m) \in G$, the left action on inputs is

$$[\mathcal{L}_g F](x, s, \theta, k) = F\big(\rho(s', \theta')^{-1}(x - t), \ (s', \theta')^{-1}(s, \theta), \ (k - m) \bmod n\big). \tag{40}$$

That is, the group index is transformed as $h \mapsto g^{-1}h$, consistent with left actions.

**Induced output action.**   For an output feature $U(x, s, \theta, k) = [F \star \Psi](x, s, \theta, k)$, the induced action is

$$[\mathcal{L}_{(t,s',\theta',m)}^c U](x, s, \theta, k) = U\Big(\rho(s', \theta')^{-1}(x - t), \ s \ominus s', \ (-1)^{s'}(\theta - \theta') \bmod 2\pi, \ (k - m) \bmod n\Big). \tag{41}$$

**Proof: Details of $E(2) \times H_n(k)$ Equivariance on Group Layer.**   We show $[\mathcal{L}_g F \star \Psi^i] = \mathcal{L}_g^c[F \star \Psi^i]$ for all $g = (t, s', \theta', m) \in G$.

$$\begin{aligned}
[\mathcal{L}_g F \star \Psi^i](x, r, \theta, k) = \sum_{y \in \mathbb{Z}^2} \sum_{s_1 \in \{0,1\}} \int_0^{2\pi} \sum_{m_1 \in \mathbb{Z}_n} \sum_{c=1}^{C^\ell} & [\mathcal{L}_g F](y, s_1, \theta_1, m_1) \\
& \cdot \ \Psi_c^i(\rho(s, \theta)^{-1}(y - x), (s, \theta)^{-1}(s_1, \theta_1), m_1 - k) \frac{d\theta_1}{2\pi} \\
= \sum_{y, s_1, m_1, c} \int_{\theta_1} & F_c(\rho(s', \theta')^{-1}(y - t), (s', \theta')^{-1}(s_1, \theta_1), m_1 - m) \\
& \cdot \ \Psi_c^i(\rho(s, \theta)^{-1}(y - x), (s, \theta)^{-1}(s_1, \theta_1), m_1 - k) \frac{d\theta_1}{2\pi}.
\end{aligned} \tag{42}$$

Let

$$
\begin{aligned}
z &= \rho(s', \theta')^{-1}(y - t), \ \Rightarrow \ y = \rho(s', \theta')z + t \\
(\tilde{s}_1, \tilde{\theta}_1) &= (s', \theta')^{-1}(s_1, \theta_1) \ \Rightarrow \ (s_1, \theta_1) = (s', \theta')(\tilde{s}_1, \tilde{\theta}_1) \\
\tilde{m}_1 &= m_1 - m \ \Rightarrow \ m_1 = \tilde{m}_1 + m.
\end{aligned}
\tag{43}
$$

Then insert all variables in Eq. 43, then

$$
[\mathcal{L}_g F \star \Psi^i](x, r, \theta, k) = \sum_{z, \tilde{s}_1, \tilde{m}_1, c} \int_{\tilde{\theta}_1} F_c(z, \tilde{s}_1, \tilde{\theta}_1, \tilde{m}_1)
$$

$$
\cdot \Psi_c^i\big( \underbrace{\rho(s, \theta)^{-1}\big(\rho(s', \theta')z + t - x\big)}_{\text{spatial rel.}}, \underbrace{(s, \theta)^{-1}\big((s', \theta')(\tilde{s}_1, \tilde{\theta}_1)\big)}_{\text{orient rel.}}, \underbrace{\tilde{m}_1 + m - k}_{\text{hue rel.}} \big)\frac{d\tilde{\theta}_1}{2\pi}
\tag{44}
$$

Then let spatial rel, orient rel. and hue rel. as follows:

$$
\begin{aligned}
\rho(s, \theta)^{-1}\big(\rho(s', \theta')z + t - x\big) &= \rho(s, \theta)^{-1}\rho(s', \theta')[z - \rho(s', \theta')^{-1}(x - t)] \\
&= \rho\big( \underbrace{(s, \theta)^{-1}(s', \theta')}_{:= q_{out}} \big)[z - \underbrace{\rho(s', \theta')^{-1}(x - t)}_{:= x^\star}] \\
(s, \theta)^{-1}\big((s', \theta')(\tilde{s}_1, \tilde{\theta}_1)\big) &= \big((s, \theta)^{-1}(s', \theta')\big)(\tilde{s}_1, \tilde{\theta}_1) = q_{out}(\tilde{s}_1, \tilde{\theta}_1) \\
\tilde{m}_1 + m - k &= \tilde{m}_1 - \underbrace{(k - m)}_{:= k^\star}.
\end{aligned}
\tag{45}
$$

Let

$$
\begin{aligned}
(q_{out})^{-1} &= \big((s, \theta)^{-1}(s', \theta')\big)^{-1} \\
&= (s', \theta')^{-1}(s, \theta) \\
&= (s', -(-1)^{s'}\theta')(s, \theta) \\
&= (s' \oplus s, -(-1)^{s'}\theta' + (-1)^{s'}\theta) \\
&= (s' \oplus s, (-1)^{s'}(\theta - \theta')) \\
&:= (s^\star, \theta^\star)
\end{aligned}
\tag{46}
$$

Then insert Eq. 45 and 46 in Eq. 44,

$$
[\mathcal{L}_g F \star \Psi^i](x, s, \theta, k) = \sum_{z, \tilde{s}_1, \tilde{m}_1, c} \int_{\tilde{\theta}_1} F_c(z, \tilde{s}_1, \tilde{\theta}_1, \tilde{m}_1) \cdot \Psi_c^i\big(\rho(q_{out}[z - x^\star], (s^\star, \theta^\star)^{-1}(\tilde{s}_1, \tilde{\theta}_1), \tilde{m}_1 - k^\star\big)\frac{d\tilde{\theta}_1}{2\pi}
$$

$$
= \sum_{z, \tilde{s}_1, \tilde{m}_1, c} \int_{\tilde{\theta}_1} F_c(z, \tilde{s}_1, \tilde{\theta}_1, \tilde{m}_1) \cdot \Psi_c^i\big(\rho(s^\star, \theta^\star)^{-1}(z - x^\star), (s^\star, \theta^\star)^{-1}(\tilde{s}_1, \tilde{\theta}_1), \tilde{m}_1 - k^\star\big)\frac{d\tilde{\theta}_1}{2\pi}
$$

$$
= [F \star \Psi^i](x^\star, s^\star, \theta^\star, k^\star) \ (\because \text{group correlation, Eq. 39})
$$

$$
= [F \star \Psi^i](\rho(s', \theta')^{-1}(x - t), s \ominus s', (-1)^{s'}(\theta - \theta'), k - m) \ (\because \text{Eq. 45 and 46})
$$

$$
= [\mathcal{L}_g^c[F \star \Psi^i]](x, s, \theta, k) \ (\because \text{definition of induced action, Eq. 41}).
\tag{47}
$$

## B.3 DETAILS OF GROUP EQUIVARIANT BATCH NORMALIZATION

When stacking JCGEL layers for large models, batch normalization is often necessary. However, a batch normalization can break equivariance (Weiler & Cesa, 2019). Motivated by Weiler & Cesa (2019), we normalize the group–indexed feature map by sharing statistics and affine parameters across the color/geometry channels. Let $X^\ell \in \mathbb{R}^{B \times C \times |H_n| \times |D_n| \times H \times W}$. For each base channel $c$,

$$
\mu_c^\ell = \frac{1}{B\,|H_n|\,|D_n|\,H\,W} \sum_{b, k, r, h, w} X_{b, c, k, r, h, w}^\ell, \quad (\sigma_c^\ell)^2 = \frac{1}{B\,|H_n|\,|D_n|\,H\,W} \sum_{b, k, r, h, w} \big(X_{b, c, k, r, h, w}^\ell - \mu_c^\ell\big)^2,
\tag{48}
$$

and we apply the fiber–shared affine map

$$\widehat{X}^\ell_{b,c,k,r,h,w} = \frac{X^\ell_{b,c,k,r,h,w} - \mu^\ell_c}{\sqrt{(\sigma^\ell_c)^2 + \varepsilon}}, \qquad Y^\ell_{b,c,k,r,h,w} = \gamma_c \, \widehat{X}^\ell_{b,c,k,r,h,w} + \beta_c.$$

Because the group action permutes only the fiber indices $(k, r)$ while $\mu^\ell_c, \sigma^\ell_c, \gamma_c, \beta_c$ are shared across them, this BN commutes with the action and thus preserves equivariance. (In practice, reshape to $(B|H_n||D_n|, C, H, W)$, apply `BatchNorm2d`, and reshape back.)

# C  DETAILS OF EQUIVARIANT

## C.1  DETAILS OF EQUIVARIANCE VALIDATION TASK EXPERIMENTAL SETTING

To validate equivariance to the hue shift and the dihedral group $D_4$, we generate 4,000 synthetic images of size $n \times n$ with $n \in \{17, 33, 65, 129\}$. Because discrete in-plane rotations on a square grid misalign the rotation center for even $n$ (causing interpolation artifacts), we restrict to odd side lengths. We compare JCGEL against a standard CNN (LeCun et al., 1998), an $E(2)$-equivariant steerable model (E2CNN) (Weiler & Cesa, 2019), and a color-equivariant convolution (CEConv) (Lengyel et al., 2023). All layers are initialized with He initialization (He et al., 2015a). For each method, we evaluate both the lifting and group layers by feeding the synthetic images and computing equivariance error as above.

## C.2  DETAILS OF EQUIVARIANT ERROR

We measure two errors, one at the lifting layer and one at the group layer:

$$\begin{aligned}
\text{Err}^{(L)} &= \text{MSE}\big(\big[\, \mathcal{L}^0_g f^0 \star \psi^0 \big], \, \big[\, \mathcal{L}^1_g [\, f^0 \star \psi^0 \,]\big]\big), \\
\text{Err}^{(G)} &= \text{MSE}\big(\big[\, \mathcal{L}^0_g f^0 \star \psi^0 \big] \star \psi^1, \, \mathcal{L}^2_g \big[\, [\, f^0 \star \psi^0 \,] \star \psi^1 \,\big]\big),
\end{aligned} \tag{49}$$

where $\mathcal{L}^\ell_g$ denotes the group action at layer $\ell$ ($\ell = 0$ for image domain, $\ell \geq 1$ for feature spaces), $f^0$ is the input image, $\psi^0$ and $\psi^1$ are the lifting and group-layer filters, and $\star$ is cross-correlation. The first line compares "transform-then-lift" versus "lift-then-transform" (lifting equivariance); the second line compares "transform-then-group-convolve" versus "group-convolve-then-transform" (group-layer equivariance).

## C.3  DETAILS OF LONG-TAILED ROTATED COLOR MNIST DATASET

**Common Experimental Setting**  We evaluate seven-layer encoders, each constructed by stacking a single convolutional primitive: standard convolution (Conv) (LeCun et al., 1998), color equivariant convolution (CEConv) (Lengyel et al., 2023), $E(2)$-equivariant steerable cnn (E2CNN) Weiler & Cesa (2019), approximately equivariant networks (AE-Net) (Wang et al., 2022), and JCGEL. All encoders are trained with the Adam optimizer (Kingma & Ba, 2015) using an initial learning rate of $10^{-4}$ and a consine-annealed schedule over 1,000 and 50 epochs with respect to the long-tailed and biased dataset. (warm up each epoch).

**Long-tailed Rotated-Color MNIST.** We construct a custom dataset from MNIST to stress-test color/geometry robustness. Each grayscale image $x \in \mathbb{R}^{28 \times 28}$ with digit label $y \in \{0, \ldots, 9\}$ is upsampled to $64 \times 64$ (bilinear) and embedded into RGB by selecting a color index $c \in \{0, 1, 2\}$ and writing the upsampled image into the $c$-th channel while zeroing the others, yielding $x' \in \mathbb{R}^{3 \times 64 \times 64}$. We then apply a rotation $R_\theta$ with $\theta = 12k°$ for $k \sim \mathcal{U}\{0, \ldots, 29\}$, producing $x'' = R_\theta(x')$. Crucially, the class label remains the original digit $y$; color and rotation act as nuisance factors (10-way classification).

To induce class imbalance in training, we draw the number of samples for each (digit, color) pair $k \in \{0, \ldots, 29\}$ from a power-law:

$$n_k \sim \Big\lceil \text{Power}(\alpha{=}0.3) \cdot N_{\max} \Big\rceil,$$

where $N_{\max}$ is the maximum per-pair budget; counts are then aggregated over color to form digit-level splits, yielding a long-tailed training set. The test set is balanced with a uniform number of examples per digit to fairly assess generalization under imbalance.

To validate robustness of JCGEL within lack of color and rotation information, we compose long-tailed rotated color MNIST (LeCun et al., 2012) dataset. Each MNIST grayscale image is upsampled to $64 \times 64$, converted to RGB by writing the image into a single channel (others zero), and then rotated in-plane by $R_\theta$ with $\theta \in \{0°, 12°, \ldots, 648°\}$ about the image center (bilinear resampling). Labels are the original digits $y$ (10 classes), independent of color/rotation. To induce a long-tailed training set, sample counts follow a power-law over (digit, color) pairs.

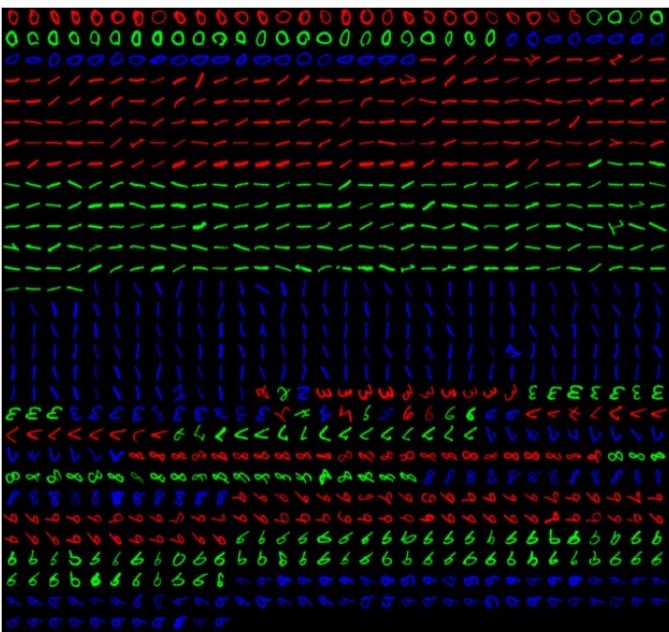

Figure 6: Rotated color MNIST long-tailed training dataset.

## C.4 DETAILS OF BIASED COLOR–ROTATION MNIST: UNIFIED SPECIFICATION

**Overview** We construct a biased MNIST variant to probe robustness against spurious correlations by coupling each digit class with preferred color and rotation. The training distribution uses two temperature (scale) parameters that control the global (inter-class) bias strength: $\tau_c$ for color and $\tau_g$ for rotation.[1] Within each class, samples are drawn from a local (intra-class) bias with a fixed sharp temperature $\tau_{\text{local}} = 0.01$ (strong concentration). The test set is constructed uniformly over all color×rotation combinations, independent of $\tau_c, \tau_g$.

**Wrapped One-Sided Exponential on a Cyclic Domain** Let items be indexed by $k \in \{0, 1, \ldots, n-1\}$ on a circle. For a center index $\mu \in \{0, \ldots, n-1\}$ and temperature $\tau > 0$, define $\lambda = \frac{1}{\tau}$ and

$$P(k \mid \mu, \tau) = \frac{\exp\big(-\lambda\,((k-\mu) \bmod n)\big)}{\sum_{i=0}^{n-1} \exp(-\lambda\,i)}. \tag{50}$$

This distribution is peaked at $k = \mu$ and decays monotonically along the cyclic order; it is not symmetric about $\mu$. Smaller $\tau$ (larger $\lambda$) yields stronger bias (sharper concentration).

**Training Set Bias** We use $N_c = 3$ colors with indices $c \in \{0{:}R, 1{:}G, 2{:}B\}$ and $N_r$ discrete rotations with indices $r \in \{0, \ldots, N_r - 1\}$. The rotation angle is $\theta(r) = \frac{360°}{N_r} r$.

**Level 1: Global (Inter-Class) Bias** Global categorical distributions are built, centered at 0 (Red and 0°):

$$P_{\text{global}}(c \mid \tau_c) = P(c \mid \mu = 0, \tau_c), \quad P_{\text{global}}(r \mid \tau_g) = P(r \mid \mu = 0, \tau_g),$$

---

[1]In code these appear as color_std and rot_std; they are not statistical standard deviations but scale (temperature) parameters.

using equation 50. For each digit class $y \in \{0, \ldots, 9\}$ we sample preferred centers

$$\mu_{c,y} \sim \text{Categorical}\big(P_{\text{global}}(c \mid \tau_c)\big), \qquad \mu_{g,y} \sim \text{Categorical}\big(P_{\text{global}}(r \mid \tau_g)\big).$$

Small $\tau_c, \tau_g$ (strong bias) cause many classes to share the same preferred pair (e.g., Red & 0°); large values diversify class-wise preferences.

**Level 2: Local (Intra-Class) Bias** Conditioned on class $y$ and its centers $(\mu_{c,y}, \mu_{g,y})$, we define class-conditional distributions with a fixed sharp temperature $\tau_{\text{local}} = 0.01$:

$$P(c \mid y) = P\big(c \mid \mu = \mu_{c,y}, \, \tau = \tau_{\text{local}}\big), \quad P(r \mid y) = P\big(r \mid \mu = \mu_{r,y}, \, \tau = \tau_{\text{local}}\big).$$

Assuming conditional independence within a class,

$$P(c, g \mid y) = P(c \mid y) \, P(g \mid y).$$

Given $N_y$ samples for class $y$, counts $\{N_{c,g}^{(y)}\}$ are drawn via

$$\{N_{c,g}^{(y)}\}_{c,g} \sim \text{Multinomial}\Big(N_y, \, \text{vec}\big(P(c, g \mid y)\big)\Big).$$

**Test Set (Uniform)** For evaluation, we allocate an equal number of samples to every triple $(y, c, r)$, yielding a uniform distribution over color×rotation per class. Implementation-wise, the per-class sample count must be divisible by $3 \times N_r$ to achieve exact uniformity (an error is raised otherwise).

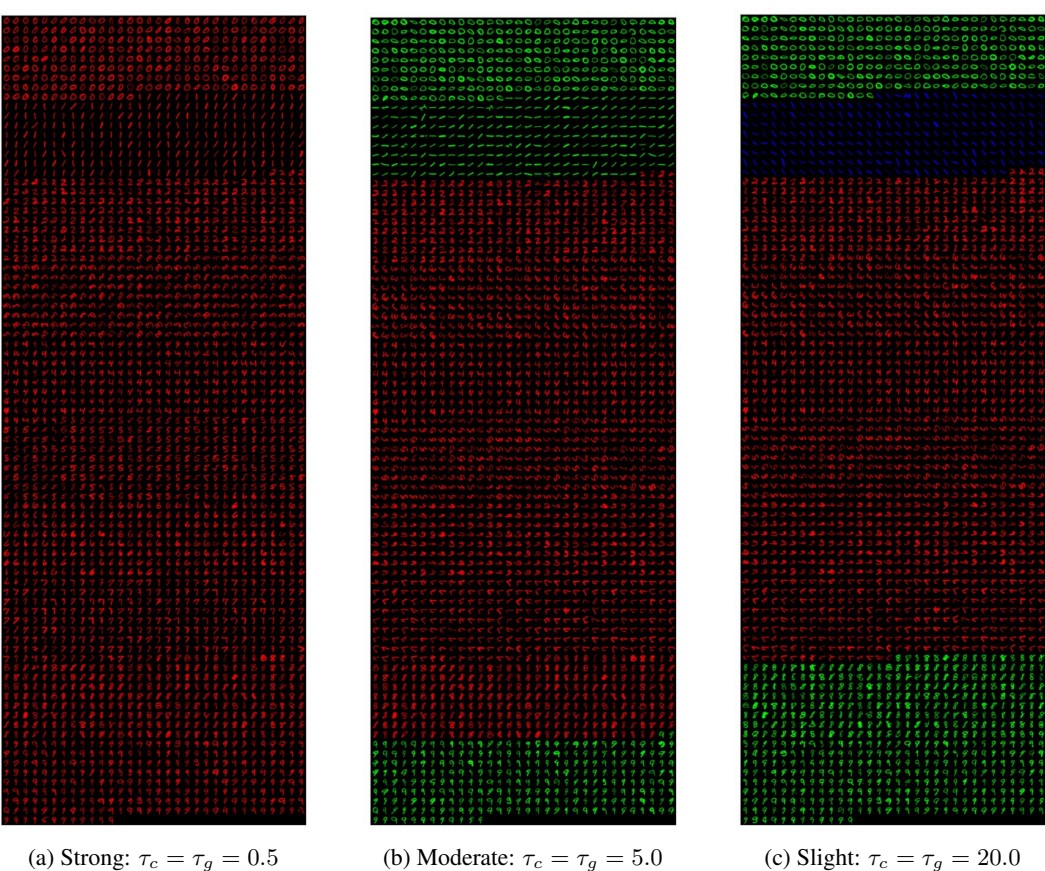

| (a) Strong: $\tau_c = \tau_g = 0.5$ | (b) Moderate: $\tau_c = \tau_g = 5.0$ | (c) Slight: $\tau_c = \tau_g = 20.0$ |

Figure 7: Rotated color MNIST biased training datasets.

### C.4.1  ⑱ ADDITIONAL LONG-TAILED TASKS (IMAGENET-LT)

**Hyper-Parameter Setting.** We set the batch size to 256, the learning rate to $10^{-4}$ with the Adam optimizer using cosine annealing over 100 epochs. We then employ baselines and the JCGEL model as follow Table 8.

**Results.** As shown in Table 6, our model outperforms the baselines, suggesting that our model is also effective on real-world datasets.

Table 6: ImageNet-LT result.

| Models | Acc. ↑ |
|---|---|
| Conv. | 30.93 |
| CEConv. | 31.19 |
| E2CNN | 31.25 |
| AE-Net | 29.22 |
| JCGEL (Ours) | **31.65** |

### C.4.2  ⑲ ADDITIONAL DISCUSSION

**Analysis of Loss-Accuracy Discrepancy in Figure 3** We address the observation regarding the discrepancy between the significant drop in Cross Entropy (CE) loss and the moderate gain in evaluation accuracy shown in Figure 3. The observed discrepancy stems from the fundamental difference between the two metrics: CE loss measures the quality of predicted probabilities (calibration), whereas accuracy depends solely on the top-1 ranking. A significant reduction in CE loss implies that our model assigns a substantially higher probability to the ground-truth class, even if this improvement is not yet sufficient to flip the top-1 prediction ranking Rahimi et al. (2020); Karandikar et al. (2021).

**Decomposition of Test Loss.** To verify this hypothesis, we decomposed the test loss into "Correctly Classified" and "Misclassified" subsets. As shown in Table 7, while the loss on correctly classified samples is comparable across methods, a striking difference appears in the misclassified samples.

Our method, JCGEL (Soft), achieves a loss of **10.087** on misclassified samples, which is significantly lower than standard baselines such as Conv (17.939) and E2CNN (18.566). This indicates that even when JCGEL fails to predict the correct class (top-1), it assigns a much higher probability mass to the true class compared to other methods (i.e., the predictions are "closer" to the truth). This improved probability assignment results in a lower overall loss despite the similar top-1 accuracy.

Table 7: Decomposition of Cross Entropy Loss on the test set. We report the average loss separately for correctly classified samples ("Correct") and misclassified samples ("Misclassified"). Lower is better.

| Method | Loss ↓ (Correct) | Loss ↓ (Misclssificed) |
|---|---|---|
| Conv. | 0.0492 | 17.9391 |
| E2CNN | 0.0582 | 18.5655 |
| CEConv | **0.0319** | 19.0237 |
| Hue-3-Sat-4 | 0.0584 | 15.1587 |
| AE-Net | 0.0759 | 33.3424 |
| JCGEL (Strict) | 0.0461 | 11.4870 |
| **JCGEL (Soft)** | 0.0491 | **10.0870** |

**Evaluation Loss Dynamics.** Regarding the increase in evaluation loss during later epochs, this is a known phenomenon when training on long-tailed distributions and evaluating on balanced sets Liu et al. (2023); Tang et al. (2020). As the model minimizes training loss by becoming overconfident on head classes (overfitting), it incurs a higher penalty on the balanced test set. However, as illustrated in Figure 3, JCGEL maintains a consistently lower evaluation loss compared to comparators throughout the training process, demonstrating superior robustness against overfitting.

### C.5  DISENTANGLEMENT LEARNING BENCHMARK DETAILS

**Experimental setting** We evaluate disentanglement on 3D Shapes (Burgess & Kim, 2018) and MPI3D (Eslami et al., 2018). For each method, we replace the VAE encoder's four convolutional layers with group-equivariant counterparts and train using Adam (learning rate $8 \times 10^{-4}$), a batch size of 512, and 500,000 training iterations. We report standard metrics—BetaVAE score (Higgins et al., 2017), FVM (Kim & Mnih, 2018), MIG (Chen et al., 2018), SAP (Kumar et al., 2018), and DCI (Eastwood & Williams, 2018).

### C.5.1  BENCHMARKS

**Setup and notation.** Let $x$ be observations generated by ground-truth factors $v = (v_1, \ldots, v_K)$. An encoder produces latent codes $z = (z_1, \ldots, z_J)$ (e.g., mean of $q_\phi(z \mid x)$). Unless stated other-

wise, latents are standardized per dimension (zero mean, unit variance over the dataset). All metrics below require access to ground-truth factors (or their labels).

$\beta$-**VAE score (FVM).** For each factor $v_k$, draw a mini-batch in which $v_k$ is held fixed while the other factors vary. Encode the batch, compute the empirical variance vector $\mathrm{Var}(z) \in \mathbb{R}^J$ across the batch, and (optionally) normalize by dataset-wide latents' variance. Train a low-capacity classifier (e.g., linear) to predict the fixed factor index $k$ from either $\mathrm{Var}(z)$ or from the index $\arg\min_j \mathrm{Var}(z_j)$. The score is the classification accuracy on held-out batches. Higher is better (one dimension is maximally insensitive when its corresponding factor is fixed). Further details are in Higgins et al. (2017).

**FactorVAE score.** Identical batching protocol as above (one factor fixed per batch), but no classifier is trained. For each batch, compute $j^\star = \arg\min_j \mathrm{Var}(z_j)$ and assign a vote that $j^\star$ corresponds to factor $k$. After collecting votes on a training stream, define a majority-vote mapping from code indices to factor indices and evaluate the accuracy on a test stream. Higher is better (same intuition as the $\beta$-VAE score, classifier-free). Further details are in Kim & Mnih (2018).

**MIG (Mutual Information Gap).** Estimate mutual information between each code and each factor, e.g., by discretizing $z_j$ and $v_k$: $I(z_j; v_k)$. For each factor $k$, sort $\{I(z_j; v_k)\}_{j=1}^J$ to get the two largest values $I_{(1),k} \geq I_{(2),k}$.

Fix a ground-truth factor index $k \in \{1, \ldots, K\}$ and consider the mutual informations

$$s_j = I(z_j; v_k), \qquad j = 1, \ldots, J.$$

Let $\pi_k$ be a permutation that sorts these scores in nonincreasing order:

$$I(z_{\pi_k(1)}; v_k) \geq I(z_{\pi_k(2)}; v_k) \geq \cdots \geq I(z_{\pi_k(J)}; v_k).$$

We then define

$$I_{(1),k} := I(z_{\pi_k(1)}; v_k) \quad \text{and} \quad I_{(2),k} := I(z_{\pi_k(2)}; v_k),$$

i.e., the largest and second-largest mutual information between any single code dimension and factor $v_k$. Consequently $I_{(1),k} \geq I_{(2),k}$ by construction.

Define

$$\mathrm{MIG} = \frac{1}{K} \sum_{k=1}^K \frac{I_{(1),k} - I_{(2),k}}{H(v_k)},$$

where $H(v_k)$ is the (discrete) entropy of factor $v_k$. Higher is better (a single code carries most of the information about each factor). Further details are in Chen et al. (2018).

**SAP (Separated Attribute Predictability).** For each factor $v_k$ and each code $z_j$, train a simple predictor from $z_j$ to $v_k$ (e.g., linear regression with $R^2$ for continuous factors or linear SVM accuracy for categorical factors), yielding scores $s_{j,k}$. For each $k$, take the gap between the top two scores: $\Delta_k = \max_j s_{j,k} - \max_{j \neq j^\star} s_{j,k}$, with $j^\star = \arg\max_j s_{j,k}$. Define $\mathrm{SAP} = \frac{1}{K} \sum_{k=1}^K \Delta_k$. Higher is better (each factor is best predicted by a unique code). Further details are in Kumar et al. (2018).

**DCI (Disentanglement–Completeness–Informativeness).** Fit a predictive model from $z$ to $v$ (e.g., gradient-boosted trees or sparse linear models) and extract nonnegative feature importances $r_{k,j}$ (importance of code $j$ for predicting factor $k$). Let $\tilde{r}_{\cdot,j}$ be importances for code $j$ normalized over factors, and $\tilde{r}_{k,\cdot}$ be importances for factor $k$ normalized over codes. Define

$$\text{Disent.} = 1 - \frac{1}{J} \sum_{j=1}^J H(\tilde{r}_{\cdot,j}), \qquad \text{Compl.} = 1 - \frac{1}{K} \sum_{k=1}^K H(\tilde{r}_{k,\cdot}),$$

where $H(\cdot)$ is the normalized entropy. Briefly, DCI-Disesnt. is the score of latent code purity: "Does each code dimension $z_j$ focus on one ground-truth factor?", and DCI-Compl. is the score of factor concentration: Is each factor $v_k$ captured mainly by one code dimension?. *Informativeness* is the predictive performance (e.g., inverse error) of the same model from $z$ to $v$. Higher disentanglement means each code is used for few factors; higher completeness means each factor is concentrated on few codes; higher informativeness means factors are predictable from $z$. Further details are in Eastwood & Williams (2018).

## C.6 CLASSIFICATION EXPERIMENTAL DETAILS

**Experimental Setting**   We report top-1 accuracy on real-world datasets (Helber et al., 2019; Krizhevsky & Hinton, 2009; Parkhi et al., 2012; Nilsback & Zisserman, 2008; Maji et al., 2013; Coates et al., 2011; Bossard et al., 2014). For each method, we replace the convolutional layers of a ResNet-18 with the candidate group equivariant operator and adjust block widths to keep parameter counts comparable across models. All models are trained with Adam for 200 epochs using a cosine-annealed learning-rate schedule (updated each epoch), following ImageNet augmentation policy, and we tune the initial learning rate over $\{10^{-3}, 10^{-4}\}$. In addition, demonstrating robustness of color and geometric variance in the real-world dataset, we randomly augmented the test samples with composite continuous hue shift and rotation. In addition, to assess robustness to color and geometric variation on real-world datasets, we apply random composite transformations at evaluation time: continuous hue shifts over the full hue circle and in-plane rotations uniformly sampled from $[0°, 360°)$.

### C.6.1 MODEL CONFIGURATIONS

Table 8: Comparison of different network architectures.

| Layer Name | Output Size | Configuration |
|---|---|---|
| **(a) Standard ResNet-18** | | |
| conv1 | $112 \times 112$ | $7 \times 7$, 64, stride 2 |
| | | $3 \times 3$ max pool, stride 2 |
| layer1 | $56 \times 56$ | $\begin{bmatrix} 1 \times 1,\ 128;\ 3 \times 3,\ 128;\ 1 \times 1,\ 512 \end{bmatrix} \times 2$ |
| layer2 | $28 \times 28$ | $\begin{bmatrix} 1 \times 1,\ 256;\ 3 \times 3,\ 256;\ 1 \times 1,\ 1024 \end{bmatrix} \times 2$ |
| layer3 | $14 \times 14$ | $\begin{bmatrix} 1 \times 1,\ 256;\ 3 \times 3,\ 256;\ 1 \times 1,\ 1024 \end{bmatrix} \times 2$ |
| layer4 | $7 \times 7$ | $\begin{bmatrix} 1 \times 1,\ 1024;\ 3 \times 3,\ 1024;\ 1 \times 1,\ 4096 \end{bmatrix} \times 2$ |
| | $1 \times 1$ | global average pool, FC(4096→classes) |
| **(b) CEConv-ResNet-18** | | |
| conv1 | $112 \times 112$ | CEConv2d $(1 \rightarrow R)$, $7 \times 7$, 64, stride 2; BN5d + ReLU |
| | $56 \times 56$ | $3 \times 3$ max pool, stride 2 (applied after merging $C \times R$) |
| layer1 | $56 \times 56$ | $\begin{bmatrix} 1 \times 1,\ 64;\ 3 \times 3,\ 64;\ 1 \times 1,\ 256 \end{bmatrix} \times 2$ |
| layer2 | $28 \times 28$ | $\begin{bmatrix} 1 \times 1,\ 128;\ 3 \times 3,\ 128;\ 1 \times 1,\ 512 \end{bmatrix} \times 2$   (first block stride 2) |
| layer3 | $14 \times 14$ | $\begin{bmatrix} 1 \times 1,\ 256;\ 3 \times 3,\ 256;\ 1 \times 1,\ 1024 \end{bmatrix} \times 2$   (first block stride 2) |
| layer4 | $7 \times 7$ | $\begin{bmatrix} 1 \times 1,\ 512;\ 3 \times 3,\ 512;\ 1 \times 1,\ 2048 \end{bmatrix} \times 2$   (first block stride 2) |
| head | $1 \times 1$ | global avg pool over $(H, W)$ on merged $C \times R$ channels; FC($2048 \times R \rightarrow$ classes) |
| **(c) E2-ResNet-18** | | |
| conv1 | $112 \times 112$ | R2Conv $7 \times 7$, to Reg$(G)$ with mult. $64$, stride 2;  IBN + ReLU |
| | $56 \times 56$ | $3 \times 3$ pointwise max pool, stride 2 |
| layer1 | $56 \times 56$ | $\begin{bmatrix} 1 \times 1,\ 64;\ 3 \times 3,\ 64;\ 1 \times 1,\ 256 \end{bmatrix} \times 2$ |
| layer2 | $28 \times 28$ | $\begin{bmatrix} 1 \times 1,\ 128;\ 3 \times 3,\ 128;\ 1 \times 1,\ 512 \end{bmatrix} \times 2$ (first block stride 2) |
| layer3 | $14 \times 14$ | $\begin{bmatrix} 1 \times 1,\ 256;\ 3 \times 3,\ 256;\ 1 \times 1,\ 1024 \end{bmatrix} \times 2$ (first block stride 2) |
| layer4 | $7 \times 7$ | $\begin{bmatrix} 1 \times 1,\ 512;\ ;\ 3 \times 3,\ 512;\ ;\ 1 \times 1,\ 2048 \end{bmatrix} \times 2$ (first block stride 2) |
| head | $1 \times 1$ | global avg pool over $(H, W)$;  FC$\big((2048) \times \gamma \rightarrow$ classes$\big)$ |
| **(d) JCGEL-ResNet-18 (Ours)** | | |
| conv1 | $112 \times 112$ | **Lifting** JCGEConv2d $(N_c : 1 \rightarrow N_c,\ N_g : 1 \rightarrow N_g)$, $7 \times 7$, 64, stride 2; CR-BN + ReLU |
| | $56 \times 56$ | **Equivariant spatial pool** $3 \times 3$, stride 2 |
| layer1 | $56 \times 56$ | $\begin{bmatrix} 1 \times 1,\ 64;\ 3 \times 3,\ 64;\ 1 \times 1,\ 256 \end{bmatrix} \times 2$ |
| layer2 | $28 \times 28$ | $\begin{bmatrix} 1 \times 1,\ 128;\ 3 \times 3,\ 128;\ 1 \times 1,\ 512 \end{bmatrix} \times 2$   (first block stride 2) |
| layer3 | $14 \times 14$ | $\begin{bmatrix} 1 \times 1,\ 256;\ 3 \times 3,\ 256;\ 1 \times 1,\ 1024 \end{bmatrix} \times 2$   (first block stride 2) |
| layer4 | $7 \times 7$ | $\begin{bmatrix} 1 \times 1,\ 512;\ 3 \times 3,\ 512;\ 1 \times 1,\ 2048 \end{bmatrix} \times 2$   (first block stride 2) |
| head | $1 \times 1$ | global average pool over $(c,\ \text{geometry},\ H,\ W)$; FC(2048 → classes) |

### C.6.2 DISCUSSION

(20) **Statistical Significance Analysis.** To rigorously assess whether the performance gains of our method are statistically meaningful rather than marginal, we conducted paired statistical tests (Student's $t$-test) between JCGEL and the strongest performing baseline (2nd-best) for each classification benchmark.

As detailed in Table 9, our method achieves statistically significant improvements ($p < 0.05$) on six out of seven datasets, including CIFAR-100 ($p = 0.029$), Oxford-Pets ($p = 0.008$), and Food-101 ($p = 0.001$). While the performance on EuroSAT remains comparable to the baseline ($p = 0.979$), the consistent statistical significance observed across the majority of standard benchmarks—as well as in our imbalanced and disentanglement experiments—confirms that the efficacy of JCGEL is robust and not attributed to random chance.

Table 9: Statistical significance test results comparing JCGEL with the second-best baseline across various datasets. The results are reported as Mean ($\pm$ Std). The $p$-values indicate the statistical significance of the improvement, with $p < 0.05$ highlighted in bold.

| Dataset | 2nd-Best | JCGEL (Ours) | $p$-value |
|---|---|---|---|
| EuroSAT | **97.83**$_{\pm 0.15}$ | 97.70$_{\pm 0.18}$ | 0.979 |
| CIFAR-100 | 77.29$_{\pm 0.01}$ | **77.51**$_{\pm 0.45}$ | **0.029** |
| Pets | 74.86$_{\pm 1.28}$ | **76.08**$_{\pm 0.80}$ | **0.008** |
| Flowers | 55.62$_{\pm 1.36}$ | **56.73**$_{\pm 1.37}$ | **0.019** |
| Aircraft | 53.02$_{\pm 0.24}$ | **54.11**$_{\pm 0.92}$ | **0.003** |
| STL-10 | 85.30$_{\pm 0.09}$ | **85.54**$_{\pm 0.26}$ | **0.011** |
| Food-101 | 81.45$_{\pm 0.31}$ | **82.62**$_{\pm 0.38}$ | **0.001** |

**Real-World Generalization via Direct-Product Discrete (Soft) Equivariance** Real-world images rarely vary along a single axis; color and geometry typically change together. Although the homogeneous space of a discrete group is smaller than that of a continuous group, our model that composes commuting discrete color and geometric actions (e.g., $(\mathbb{Z}^2 \rtimes D_4) \times H_n$) consistently improves performance over E2CNN across diverse vision tasks. Moreover, JCGEL surpasses JCGEL-C, which is equivariant to $SE(2) \times H_n$ as shown in Table 10, suggesting that a direct product of discrete groups can be an effective choice for real-world generalization.

Table 10: Discrete vs. continuous group equivariant model. JCGEL-G denotes equivariant to $SE(2) \times H_n(k)$ model.

| | JCGEL | JCGEL-C |
|---|---|---|
| EuroSAT | **97.70**($\pm 0.18$) | 97.52($\pm 0.18$) |
| Aircraft | **54.11**($\pm 0.92$) | 52.95($\pm 0.87$) |
| STL10 | **85.54**($\pm 0.26$) | 85.29($\pm 0.57$) |

Two practical considerations support this finding. First, in real-world pipelines, continuous transformations act through data augmentation on the image grid, and this effectively broadens the coverage achieved by a discrete product group and enables JCGEL to generalize to many unseen poses and hues. Second, continuous group strict equivariant models assume ideal group actions that may conflict with common augmentations (e.g., rotated images leave empty regions that are padded, which is not a true group action). This mismatch affects strict formulations, and even soft methods that target continuous groups impose stronger constraints than discrete equivariance, which can hinder performance under non-ideal image-domain operations. In summary, a direct-product discrete formulation is well aligned with real-world conditions, explaining why JCGEL tends to achieve higher accuracy and robustness across varied environments.

## D (21) LIMITATIONS AND FUTURE WORK

**Computational Overhead.** The primary limitation of our proposed method lies in its computational cost compared to baselines. Operating on the high-dimensional product group $(\mathbb{Z}^2 \rtimes G_o) \times H_n$ inevitably expands the feature space, leading to increased computational complexity and memory usage.

As summarized in Table 11, relative to the standard convolutional baseline ($\times 1.00$), our JCGEL layer exhibits lower training throughput ($\approx \times 0.14$) and significantly higher peak GPU memory (VRAM) consumption ($\approx \times 4.37$). This overhead is inherent to the explicit construction of joint equivariance across color and geometric transformations.

We acknowledge this trade-off between computational efficiency and model robustness. While standard efficient models offer faster inference, they fail to achieve the superior performance and stability demonstrated by our method (as shown in Tables 2, 4, and 6). Consequently, reducing the computational burden of product-group convolutions—potentially through sparse group operations or approximation techniques—remains a critical direction for our future optimization.

Table 11: Comparison of computational efficiency normalized to the standard CNN baseline ($\times 1.00$). Throughput is measured in images/sec (higher is better), and VRAM usage is measured in GB (lower is better).

| Method | Training Speed ($\uparrow$) | Inference Speed ($\uparrow$) | VRAM Usage ($\downarrow$) |
|---|---|---|---|
| Conv (Baseline) | $\times 1.00$ | $\times 1.00$ | $\times 1.00$ |
| CEConv | $\times 0.84$ | $\times 0.72$ | $\times 1.37$ |
| E2CNN | $\times 0.31$ | $\times 0.27$ | $\times 1.74$ |
| Hue-4-Sat-3 | $\times 0.12$ | $\times 0.09$ | $\times 2.04$ |
| **JCGEL (Ours)** | $\times 0.14$ | $\times 0.09$ | $\times 4.37$ |