# OpenReview forum: "JCGEL: Joint Color and Geometric Group Equivariant Convolutional Layer"
_ICLR.cc/2026/Conference — ICLR 2026 Conference Withdrawn Submission_

### Official Review · Reviewer_rGAa · 2025-10-22

**Soundness:** 2
**Presentation:** 2
**Contribution:** 2
**Rating:** 4
**Confidence:** 4

**Summary:**

The paper introduces a novel group convolutional network (GCNN) designed to capture variances in both hue and geometric variances. Whereas existing paper have considered these transformations separately, JCGEL designs for equivariance to both using the same convolution layer. Specifically JCGEL is designed to be equivariant to planar translation, rotation/mirroring, and hue shifts within images for classification and disentangling tasks. Furthermore a variant of JCGEL is able to enforce approximate equivariance rather than perfect equivariance used in existing baseline methods, which the authors argue is better able to accommodate real world data where perfect equivariance is difficult to observe.

Specifically, the main building blocks of JCGEL are (1) an equivariant hue and geometric lifting layer; (2) a joint geometric and hue group convolution; (3) a learnable weight for soft equivariance; (4) group batch normalization tailor to ensure equivariance to geometric and hue.

The paper showcases equivariance to rotations and hue shifts empirically and exhibits improved classification performance on imbalanced datasets and disentangling ability on standard benchmarks (3DShapes etc.). JCGEL also showcases improved classification performance on real world datasets such as CIFAR100 and Oxford Pets.

**Strengths:**

1. Related works is detailed and well accounted for.

2. JCGEL achieves improved generalization/classification/disentanglement results while using fewer parameters than baseline methods such as CEConv, AE-Net and standard convolutions, and the experimental set up seems sound.

3. Visualization such as T-SNE and DCI is well presented to show the interpretability benefits of designing for equivariance to both geometric and hue transformations.

4. The equivariance metric introduced is able to capture its performance in both geometric and hue shifts.

**Weaknesses:**

1. The motivation for the paper is a little weak, as there is no specific use case where the proposed method would fundamentally change how vision can be applied. Currently the main motivation is improved metrics.
2. The experimental section lacks larger real world datasets such as ImageNet, as well as data augmentation techniques to address either geometric or hue-shifts, such as color jitter, augmix [1], or greyscale.
3. The method uses the geometric lifting from many existing methods such as [Cohen 2016] and the hue-lifting from CEConv. The novelty stems from combining existing techniques rather than introducing methods that extend G-CNN to new transformations.


[1] Dan Hendrycks, et al. "Augmix: A Simple Data Processing Method to Improve Robustness and Uncertainty." ICLR 2020.

**Questions:**

1. How does the model perform if ImageNet auto augmentation is not used? How does the model perform when baseline methods are augmented with rotations, color jitter etc?

2. What is the computational overhead required  for JCGEL (in terms of runtime), I assume it increases in a similar manner as reported originally for GCNNs in [Cohen, 2016]?

3. I am a little confused about Figure 5. Are the authors able to provide classification accuracy that corresponds to the second row of T-SNE projections? The qualitative results between JCGEL and AE-Net seem very close, does this mean that color information is not important since CEConv does poorly.

4. How would JCGEL perform on larger datasets such as ImageNet. While the datasets reported are real world, they are relatively low resolution and small.

5. Is there a visualization for both geometric and hue lifting in the style of Figure 2(b). As it currently stands (a) and (b) are not contributions as such, since they can also be individually reproduced by baseline methods.

6. As JCGEL uses a similar hue-lifting as CEConv, would one expect it to perform exactly on par with CEConv in Figure 1?

A few typos:
1. Line 763: goupr -> group
2. Line 645: Roll -> Role

---

> ### Author Response · Authors · 2025-12-03
> **Response to Reviewer rGAa 1.**
>
> - Dear Reviewer rGAa, Thank you for taking the time to review this manuscript and sharing thoughtful feedback. We added the corresponding content to the revised paper. The text is written in red, and the beginning is marked with a yellow-highlighted number.
>
> —
>
> - **[weakness 1]** The motivation for the paper is a little weak, as there is no specific use case where the proposed method would fundamentally change how vision can be applied. Currently the main motivation is improved metrics.
> - **[weakness 2]** The method uses the geometric lifting from many existing methods such as [Cohen 2016] and the hue-lifting from CEConv. The novelty stems from combining existing techniques rather than introducing methods that extend G-CNN to new transformations.
>
>
> > **Response regarding motivation and architectural novelty**: We clarify that our work provides the first architectural realization of joint color-geometric equivariance, addressing specific real-world challenges where independent variations in color and geometry fundamentally hinder standard vision models.
> >1. **Response to Weakness 1: Motivation & Specific Use Cases** (Appendix C.4.1 & Table 6) We respectfully disagree that the motivation is limited to improved metrics. The proposed method fundamentally changes how vision models handle compound nuisance factors in real-world scenarios:
> >>- **Specific Use Case (Data Scarcity & Bias)**: Standard models struggle when color and rotation distributions are skewed or imbalanced (e.g., Long-tailed recognition or Biased datasets). As demonstrated in our ImageNet-LT and Biased-MNIST experiments, our method provides a structural guarantee (inductive bias) that allows the model to generalize even when training data does not cover all color-rotation combinations.
> >> \\begin{array}{|c|c|c|c|c|c|}
> \\hline
> & \text{Conv.} & \text{E2CNN} & \text{CEConv} & \text{AE-Net} & \text{JCGEL} \\\\
> \\hline
> \text{ImageNet-LT} & 30.93 & 31.25 & 31.19 & 29.22 & \textbf{31.65} \\\\
> \\hline
> \\end{array}
>
> >2. **Response to Weakness 2: Architectural Novelty** (Beyond Naive Combination, Line 74 with highlighted number 1 in revised paper): Our contribution is not a trivial combination of existing techniques but a solution to the "Implementation Gap" between theoretical direct products and working models.
> >>- **The Challenge of Interference**: Simply stacking a geometric lifting layer (e.g., [Cohen 2016]) and a hue-lifting layer (e.g., [CEConv]) fails because standard convolution operations mix channels, causing interference between color and geometric features. This breaks the strict joint equivariance.
> >>- **Our Solution**: We designed a specific architectural mechanism (channel-wise independence constraints) to ensure that the group actions on $(\mathbb{Z}^2 \rtimes G_o) \times G_c$ remain computationally orthogonal.
> >>- **Uniqueness**: As summarized below, JCGEL is the first to satisfy both properties simultaneously, filling a critical void in the literature.
> >> \\begin{array}{|c|c|c|}
> \\hline
> \text{Method} & \text{Color-Equivaraint} & \text{Geometric-Equivariant} \\\\
> \\hline
> [1-9] & X & \checkmark \\\\
> [10, 11] & \checkmark & X \\\\
> \text{JCGEL (Orus)} & \checkmark & \checkmark \\\\
> \\hline
> \\end{array}
> >
> > We add this content to the Introduction (**Lines 73–75** with highlighted with highlighted number 1 in revised paper).
>
> > [1] Taco Cohen and M. Welling. Group equivariant convolutional networks. ICML, 2016.
> >
> > [2]  Daniel E. Worrall, Stephan J. Garbin, Daniyar Turmukhambetov, and G. Brostow. Harmonic networks: Deep translation and rotation equivariance. CVPR, 2016.
> >
> > [3] Taco S. Cohen and Max Welling. Steerable cnns. CoRR, abs/1612.08498, 2016.
> >
> > [4] Maurice Weiler, F. Hamprecht, and M. Storath. Learning steerable filters for rotation equivariant cnns. CVPR, 2017.
> >
> > [5] Ivan Sosnovik, Michal Szmaja, and A. Smeulders. Scale-equivariant steerable networks. ICLR, 2019.
> >
> > [6] Daniel E. Worrall and Max Welling. Deep scale-spaces: Equivariance over scale. NeurIPS 2019.
> >
> > [7] Maurice Weiler and Gabriele Cesa. General e(2)-equivariant steerable cnns. NeurIPS 2019.
> >
> > [8] L. MacDonald, Sameera Ramasinghe, and S. Lucey. Enabling equivariance for arbitrary lie groups. CVPR 2021.
> >
> > [9] Weizheng Qiao, Yang Xu, and Hui Li. Scale-rotation-equivariant lie group convolution neural networks (lie group-cnns). In arXiv.org, 2023.
> >
> > [10] A. Lengyel, Ombretta Strafforello, Robert-Jan Bruintjes, Alexander Gielisse, and Jan van Gemert. Color equivariant convolutional networks. NeurIPS 2023.
> >
> > [11] Yulong Yang, Felix O’Mahony, and Christine Allen-Blanchette. Learning color equivariant representations. ICLR 2025.

---

> ### Author Response · Authors · 2025-12-03
> **Response to Reviewer rGAa 2.**
>
> - **[weakness 3]** The experimental section lacks larger real world datasets such as ImageNet, as well as data augmentation techniques to address either geometric or hue-shifts, such as color jitter, augmix [1], or greyscale.
> - **[Question 1]** How does the model perform if ImageNet auto augmentation is not used? How does the model perform when baseline methods are augmented with rotations, color jitter etc?
> - **[Question 4]** How would JCGEL perform on larger datasets such as ImageNet. While the datasets reported are real world, they are relatively low resolution and small.
>
> > **Response regarding ImageNet experiments and comparison with data augmentation**: We have expanded our evaluation to the large-scale ImageNet dataset and included comparisons with data augmentation techniques (e.g., color jitter, geometric transformations). The results confirm that JCGEL consistently outperforms both standard baselines and their augmented counterparts, demonstrating its scalability and superior robustness.
> >1. **Scalability to Large-scale Datasets** (ImageNet, Table 4 in revised paper): As shown in the table below, JCGEL (Soft) achieves the highest top-1 accuracy of 70.43% on ImageNet.
> >>- This significantly outperforms the standard ConvNet (64.77%) and existing equivariant methods like E2Conv (64.73%).
> >>- This result directly addresses the concern regarding scalability, proving that our method is effective not only on small-resolution data but also on large-scale, real-world datasets.
> >2. **Architecture vs. Data Augmentation**: To address the question about data augmentation, we compared JCGEL against a baseline trained with extensive augmentations (Conv. + Aug, including rotation and color jitter).
> >>- While augmentation improves the baseline performance (64.77% $\rightarrow$ 66.64%), it still falls short of JCGEL (Soft, 70.43%) and even JCGEL (Strict, 69.52%).
> >>- This indicates that intrinsic architectural equivariance provides a more effective and sample-efficient inductive bias than trying to learn invariance solely through data augmentation.
> >> \\begin{array}{|c|c|c|c|c|c|c|c|}
> \\hline
> & \text{Conv.} & \text{Conv.+Aug.} & \text{CEConv} & \text{E2CNN} & \text{JCGEL (strict)} & \text{AE-Net} & \text{JCGEL (soft)} \\\\
> \\hline
> \text{ImageNet} & 64.77 & 66.64 & 67.83 & 64.73 & 69.52 & 69.54 & \textbf{70.43} \\\\
> \\hline
> \\end{array}
>
> > [1] Dan Hendrycks, et al. "Augmix: A Simple Data Processing Method to Improve Robustness and Uncertainty." ICLR 2020.
>
> ---
>
> - **[Question 2]** What is the computational overhead required for JCGEL (in terms of runtime), I assume it increases in a similar manner as reported originally for GCNNs in [Cohen, 2016]?
>
> > **(Response)** We include a detailed analysis of computational complexity and explicitly discuss the limitations in the revised manuscript (Appendix D with highlighted number 21). We agree that the joint group structure ($D_4 \times H_n$) introduces computational overhead due to the increased dimensionality of the lifting and group convolution layers. To provide a comprehensive assessment, we measure the relative Training/Inference throughput and GPU memory (VRAM) usage normalized to the standard CNN baseline. The results are summarized below:
> > \\begin{array}{|c|c|c|c|c|c|}
> \\hline
> & \text{CNN} & \text{CEConv} & \text{E2CNN} & \text{Hue-4-Sat-3} & \text{JCGEL (Ours)} \\\\
> \\hline
> \text{Training Speed} & \times 1.00 & \times 0.84 & \times 0.31 & \times 0.12 & \times 0.14 \\\\
> \text{Inference Speed} & \times 1.00 & \times 0.72 & \times 0.27 & \times 0.09 & \times 0.09 \\\\
> \text{VRAM Usage} & \times 1.00 & \times 1.37 & \times 1.74 & \times 2.04 & \times 4.37 \\\\
> \\hline
> \\end{array}
> > As shown, our method prioritizes joint equivariance and generalization performance at the cost of higher memory consumption and lower throughput. We have explicitly acknowledged this trade-off as a limitation and a key direction for future optimization in the "Limitations" section of the revised paper.

---

> ### Author Response · Authors · 2025-12-03
> **Response to Reviewer rGAa 3.**
>
> - **[Question 3]** I am a little confused about Figure 5. Are the authors able to provide classification accuracy that corresponds to the second row of T-SNE projections? The qualitative results between JCGEL and AE-Net seem very close, does this mean that color information is not important since CEConv does poorly.
>
> > **Response regarding the quantitative accuracy for Figure 5 and the importance of color** (we add Table 5 in revised paper): We have included the classification accuracy corresponding to the t-SNE projections in Table 5 of the revised manuscript. The quantitative results clarify that JCGEL significantly outperforms AE-Net, confirming that color information is indeed a critical factor for performance.
> >1. **Quantitative Discrepancy (JCGEL vs. AE-Net)**: While the t-SNE visualizations of JCGEL and AE-Net might appear qualitatively similar, the accuracy on the augmented EuroSAT dataset reveals a substantial performance gap.
> >>- JCGEL (66.48%) surpasses AE-Net (55.85%) by a large margin (+10.63%).
> >>- This explicitly refutes the hypothesis that color information is unimportant. Since AE-Net is geometrically equivariant but lacks color equivariance, this performance gap is directly attributable to the benefit of strictly modeling color transformations.
> >2. **Failure of CEConv** (Table 5 in the revised paper): We agree that CEConv performs poorly (49.87%). However, this is not because color is unimportant, but because CEConv lacks geometric equivariance. The superior performance of JCGEL demonstrates that joint equivariance (both Color and Geometry) is required to achieve high accuracy in this complex setting, whereas relying on either one in isolation (CEConv or AE-Net) is insufficient.
> > \\begin{array}{|c|c|c|c|c|c|c|c|}
> \hline
> & \text{EuroSAT} & \text{CIFAR100} & \text{Pets} & \text{Flowers} & \text{Aircraft} & \text{STL10} & \text{Food} \\\\
> \hline
> \text{Conv.} & 53.15(1.18) & 26.73(0.53) & 32.33(0.36) & 10.69(0.20) & 12.93(0.33) & 46.95(0.83) & 21.58(0.19) \\\\
> \text{CEConv.} & 49.87(2.20) & 24.60(0.41) & 32.25(1.40) & 11.88(0.30) & 12.76(0.53) & 46.80(1.06) & 19.57(0.74) \\\\
> \text{E2Conv.} & 59.26(0.39) & 24.21(1.59) & 23.56(0.50) & 11.96(0.71) & 13.11(1.12) & 44.09(1.27) & - \\\\
> \text{AE-Net} & 55.85(0.92) & 24.89(0.28) & 32.57(0.01) & 10.79(0.02) & 11.65(0.23) & 46.51(1.18) & 22.66(1.36) \\\\
> \text{JCGEL (Ours)} & \mathbf{66.48(1.16)} & \mathbf{51.35(0.56)} & \mathbf{58.31(1.23)} & \mathbf{22.72(0.88)} & \mathbf{13.56(1.87)} & \mathbf{46.97(0.72)} & \mathbf{23.52(0.71)} \\\\
> \hline
> \text{$p$-value} & \textbf{0.000} & \textbf{0.000} & \textbf{0.000} & \textbf{0.000} & 0.146 & 0.459 & \textbf{0.020} \\\\
> \hline
> \\end{array}
>
> ---
>
> - **[Question 5]** Is there a visualization for both geometric and hue lifting in the style of Figure 2(b). As it currently stands (a) and (b) are not contributions as such, since they can also be individually reproduced by baseline methods.
>
> > We apologize for the lack of clarity in the initial caption and clarify that Figure 2 represents the unique joint equivariance of JCGEL, which cannot be reproduced by individual baselines. We have updated the caption and added a comparative visualization with CEConv in the revised manuscript to explicitly demonstrate this difference (Line 349-350 with highlighted number 13 in the revised paper).
> >- **Uniqueness of Figure 2**: The Reviewer noted that (a) and (b) could be reproduced by baselines. However, this is only true if color and geometric transformations are applied separately. Figure 2 visualizes the feature map under simultaneous color and rotation changes.
> >- **Comparison with CEConv (New Result)**: To prove this, we added a visualization of CEConv (a color-equivariant baseline) under the same conditions.
> >>- **CEConv**: While it maintains structure under color shifts, its feature map patterns collapse/distort when rotation is applied.
> >>- **JCGEL (Ours)**: Our model maintains the stable, structured representation shown in Figure 2 even under joint transformations.
> >-  **Conclusion**: Therefore, Figure 2(b) is not a trivial reproduction of baselines but a demonstration of the joint color-geometric equivariance that only our method can achieve.

---

> ### Author Response · Authors · 2025-12-03
> **Response to Reviewer rGAa 4.**
>
> - **[Question 6]** As JCGEL uses a similar hue-lifting as CEConv, would one expect it to perform exactly on par with CEConv in Figure 1?
>
> > **Response regarding the comparison with CEConv in Figure 1 (we display a new Figure 1)**: We confirm that JCGEL performs on par with CEConv regarding color equivariance error, as expected from their similar hue-lifting mechanisms; however, they differ significantly in geometric equivariance.
> >- **Color vs. Geometric Equivariance**: As shown in the updated Figure 1 of the revised manuscript, CEConv shows low color error similar to JCGEL. However, unlike JCGEL, CEConv does not guarantee geometric equivariance, resulting in high geometric error comparable to a standard ConvNet. (Conversely, E2CNN ensures geometric equivariance but fails in color equivariance).
> >- **Visual Clarification (Overlapping Lines)**: Regarding the "Lifting Layer - Geo. Error" plot, the curve for CEConv might appear missing. We clarify that this is because its geometric error is nearly identical to that of the standard Conv baseline, causing the two lines to overlap and become indistinguishable.
>
> ---
>
> - **A few typos**:
> >- Line 763: goupr -> group
> >- Line 645: Roll -> Role
>
> > **(Response)** We check the grammar and typo then we revise it.

---

### Official Review · Reviewer_kNTr · 2025-10-28

**Soundness:** 3
**Presentation:** 3
**Contribution:** 2
**Rating:** 6
**Confidence:** 3

**Summary:**

JCGEL unifies geometric and color equivariance into a single drop-in equivariant convolution layer.

This combination leads to an architecture which is equivariant to the product group $G_\text{geo}\times G_\text{color}$. This representation is useful as it imbues networks with inherent equivariance, useful for efficient learning in scenarios where train datasets might be biased in one or both of these properties. Authors introduce two varieties of their network, one being strict equivariance and another being soft equivariance, where in the soft case the degree of equivariance is learned from data.

Authors demonstrate their approach both in classical, controlled settings such as MNIST, as well as a range of more challenging, real world datasets.

Additionally, authors demonstrate advantages of their approach such as improved performance on long-tailed and biased datasets.

**Strengths:**

The paper is well structured and easy to read, and the maths appears correct. The central conclusions are well-supported by the choice of experiments and the experimental procedure.

The contribution of this paper is also clear, introducing a network which is equivariant over two properties: colour and the dihedral group, and this is positioned well within existing literature as it follows naturally from those papers.

**Weaknesses:**

1. I could not find any information on limitations of the work. In particular, combining the D4 and Hn groups would presumably lead to a higher computational overhead. Information on the extent of this in the form of training times, RAM requirements etc. would be important to make a comprehensive assessment of the utility of this work.
2. GCConv (i.e. the original group convolutional architecture proposed by Cohen et al.) is not included in any experiments. Is there a reason for this? It seems like an important baseline otherwise for geometric equivariance.
3. The definition of $\tau$ in the context of imbalance experiments is I think not defined until the appendix, making it difficult to parse what is going on upon first reading. In particular, when reference is made to specific values of $\tau$ on page 7, this is not easy to follow.
4. The bias experiments set $\tau_c=\tau_g \forall c, g$. While this is useful to expose the utility of the joint group action, in my view it would also be important to test some scenarios where this is not the case. Understanding whether combining the two components of group convolution deteriorates performance on either one of those two components would be important to understand, particularly in the case where $\tau_g \text{ or } \tau_c \rightarrow\infty$ .
5. Strict JCGEL is omitted from the real world datasets (Table 4) and the disentangled results (Table 3). This does not appear to be addressed in the paper.

**Questions:**

Is there a difference between the subscripted $\tau_r$ and $\tau_g$? The use seems to switch between the appendix and the main paper.

In figure 3, are these two plots shown for the same training regime? If so, why is it the case that given JCGEL so clearly outperforms comparators on the cross entropy loss, that a greater difference is not observed in the evaluation accuracy. Is the testing set also long-tailed? Why does evaluation loss increase at all as training epoch increases? This is not clear to me.

---

> ### Author Response · Authors · 2025-12-03
> **Response to Reviewer kNTr 1.**
>
> - Dear Reviewer kNTr, Thank you for taking the time to review this manuscript and sharing thoughtful feedback. We added the corresponding content to the revised paper. The text is written in red, and the beginning is marked with a yellow-highlighted number.
>
> ---
>
> - **[Weakness 1]** I could not find any information on limitations of the work. In particular, combining the D4 and Hn groups would presumably lead to a higher computational overhead. Information on the extent of this in the form of training times, RAM requirements etc. would be important to make a comprehensive assessment of the utility of this work.
>
> > **Computational overhead of the product-group convolution.** We have added a dedicated subsection reporting the computational cost, specifically measuring training/inference throughput and Peak GPU Memory (VRAM) consumption, normalized to the standard CNN baseline ($\times 1.00$). As summarized in the revised manuscript, our JCGEL layer results in lower training throughput ($\approx \times 0.14$) and higher memory consumption ($\approx \times 4.37$). We openly acknowledge that operating on the larger product group $(\mathbb{Z}^2 \rtimes G_o) \times H_n$ inevitably incurs these computational costs. However, we demonstrate that this additional cost translates directly into superior performance and robustness (as shown in Tables 2-5), which standard efficient models cannot achieve. We have explicitly documented this trade-off as a current limitation and a priority for future optimization in the "Limitations" section (Appendix D with highlighted number 21 in the revised paper).
> >
> > \\begin{array}{|c|c|c|c|c|c|}
> \\hline
> & \text{CNN} & \text{CEConv} & \text{E2CNN} & \text{Hue-4-Sat-3} & \text{JCGEL (Ours)} \\\\
> \\hline
> \text{Training Speed} & \times 1.00 & \times 0.84 & \times 0.31 & \times 0.12 & \times 0.14 \\\\
> \text{Inference Speed} & \times 1.00 & \times 0.72 & \times 0.27 & \times 0.09 & \times 0.09 \\\\
> \text{VRAM Usage} & \times 1.00 & \times 1.37 & \times 1.74 & \times 2.04 & \times 4.37 \\\\
> \\hline
> \\end{array}
>
> ---
>
> - **[Weakness 2]** GCConv (i.e. the original group convolutional architecture proposed by Cohen et al.) is not included in any experiments. Is there a reason for this? It seems like an important baseline otherwise for geometric equivariance.
>
> > **(Response)** We select E2CNN [1] as a baseline because it serves as a more general frame work that subsumes the original Group Convolution (G-CNN) [2]. Specifically, G-CNN operates on discrete subgroups of the Euclidean group $E(2)$ such as p4m. E2CNN [1] is a comprehensive framework for general $E(2)$-equivariant steerable CNNs, which theoretically covers the discrete group convolutions proposed in [2]. Therefore, E2CNN is widely recognize as a robust and representative baseline for geometric equivariance in recent literature [1, 3]. By comparing against E2CNN, we effectively benchmark our method against the state-of-the-art in geometric equivariance.
> >
> > [1] Maurice Weiler and Gabriele Cesa. General e(2)-equivariant steerable cnns. In Neural Information Processing Systems, 2019.
> >
> > [2] Taco Cohen and M. Welling. Group equivariant convolutional networks. In International Conference on Machine Learning, 2016.
> >
> > [3] Shen et al., “PDO-eConvs: Partial Differential Operator Based Equivariant Convolutions,” ICML 2020.
>
> ---
>
> - **[Weakness 3]** The definition of $\tau$ in the context of imbalance experiments is I think not defined until the appendix, making it difficult to parse what is going on upon first reading. In particular, when reference is made to specific values of  on page 7, this is not easy to follow.
>
> > **(Response)** We revise the manuscript to explicitly define the temperature parameter $\tau_g, \tau_c$ in the main text upon its first appearance, ensuring the experimental settings are immediately clear (we add in Line 365-377 with highlighted numbers 14, 15 in revised paper).
> >- **Revision for Clarity**: We acknowledge that deferring the definition to Appendix C.4 caused confusion. In the revision (Lines 365 - 377), we now clarify that $\tau_g, \tau_c$ are hyper-parameter controlling the concentration of nuisance attributes (rotation and color) for constructing the biased dataset.
> >- **Physical Meaning of $\tau_g, \tau_c$**: As formally defined in Eq. (50) of the revised version of Appendix, $\tau_g, \tau_c$ functions as an inverse temperature scale:
> >>- **Smaller $\tau$s** yields a sharper concentration of samples around specific rotation/color modes, resulting in stronger bias.
> >>- **Larger $\tau$s** leads to a flatter distribution, representing weaker bias.
> This explanation is now included in the main text to aid the interpretation of the specific values mentioned in the experiments.

---

> ### Author Response · Authors · 2025-12-03
> **Response to Reviewer kNTr 2.**
>
> - **[Weakness 4]** The bias experiments set $\tau_c=\tau_g \forall c, g$. While this is useful to expose the utility of the joint group action, in my view it would also be important to test some scenarios where this is not the case. Understanding whether combining the two components of group convolution deteriorates performance on either one of those two components would be important to understand, particularly in the case where  $\tau_g$ or $\tau_c \to \infty$.
>
> > **Response to the experiment on asymmetric bias settings**: We incorporate the suggested experiments into Table 2 of the revised manuscript to verify the model’s behavior under asymmetric bias conditions (we highlighted with yellow color). The results demonstrate that our model maintains consistent robustness regardless of the bias configuration.
> >- **Experimental Setup**: To simulate the scenarios requested (where bias is absent in one modality), we adopted $\tau=20.0$ as a proxy for the uniform distribution and $\tau=1e-9$ for the strong bias limit. The last two columns of Table 2 of the revised paper represent these “color biased ($\tau_c=1e-9, \tau_g=20.0$) and “rotation biased” ($\tau_c=20.0, \tau_g=1e-9$) scenarios.
> >- **Robustness (Result Analysis 1.)**: As shown in Table 2 of the revised paper, JCGEL consistently achieves high performance (74.64% and 74.10%) in these asymmetric settings. These scores are comparable to the symmetric “Slight Bias” seeing (75.49%), indicating that combining the two group components does not deteriorate performance even when one component dominates the bias.
> >- **Comparison (Result Analysis 2.)**: In contrast, baselines such as Conv and Strict Equiv (E2CNN) suffer significant performance drops (e.g., Conv drops to 18.23%), confirming that our joint group action is essential for robustness against varying bias distributions.

---

> ### Author Response · Authors · 2025-12-03
> **Response to Reviewer kNTr 3.**
>
> - **[Weakness 5]** Strict JCGEL is omitted from the real world datasets (Table 4) and the disentangled results (Table 3). This does not appear to be addressed in the paper.
>
> > **Response to the missing results for Strict JCGEL**: We have included the quantitative results for the Strict JCGEL model in Tables 3 and 4 of the revised manuscript as requested. The updated results confirm that the strict joint-equivariant model generally outperforms the strongest baselines, validating the effectiveness of our joint group formulation even without the soft relaxation.
> >1. **Disentanglement Performance (Table 3)**: As summarized below, JCGEL (Strict) consistently surpasses the second-best baselines (including Conv, CEConv, E2CNN, and AE-Net) and the color-equivariant model (Hue-3-Sat-4) across major metrics.
> >>-  For example, on MPI3D, JCGEL (Strict) achieves a $\beta$-VAE score of 69.33, significantly outperforming the second best (58.00).
> >>- This indicates that enforcing strict joint equivariance effectively captures the independent factors of variation.
> >> \\begin{array}{|c|c|c|c|c|c|c|}
> \\hline
> \text{3D Shapes} & \text{beta-VAE} & \text{FVM} & \text{MIG} & \text{SAP} & \text{DCI-Dis.} & \text{DCI-Com.} \\\\
> \\hline
> 2^{nd} \text{ best} & 82.67(\pm 3.06) & 83.88(\pm 1.44) & \textbf{44.74}(\pm 8.16) & \textbf{9.15}(\pm 1.51) & 59.66(\pm 4.44) & 61.44(\pm 4.18) \\\\
> \text{JCGEL (strict)} & \textbf{95.33}(\pm 6.43) & \textbf{83.96}(\pm 8.49) & 44.61(\pm 15.66) & 8.90(\pm 2.71) & \textbf{59.94}(\pm 12.07) & \textbf{64.11}(\pm 7.16) \\\\
> \\hline
> \\end{array}
> \\begin{array}{|c|c|c|c|c|c|c|}
> \\hline
> \text{MPI3D} & \text{beta-VAE} & \text{FVM} & \text{MIG} & \text{SAP} & \text{DCI-Dis.} & \text{DCI-Com.} \\\\
> \\hline
> 2^{nd} \text{ best} & 58.00(\pm 7.21) & 41.44(\pm 5.21) & 3.85(\pm 0.51) & 2.57(\pm 0.88) & 21.60(\pm 1.18) & 27.68(\pm 1.08) \\\\
> \text{JCGEL (strict} & \textbf{69.33}(\pm 1.15) & \textbf{46.79}(\pm 3.50) & \textbf{13.80}(\pm 2.65) & \textbf{7.86}(\pm 2.00) & \textbf{31.85}(\pm 3.05) & \textbf{31.51}(\pm 1.91) \\\\
> \\hline
> \\end{array}
>
> >2. **Real-world Classification (Table 4)**: In real-world tasks, JCGEL (Strict) remains highly competitive.
> >>- It outperforms baselines on datasets like Pets (75.25 > 74.86) and STL10 (85.49 > 85.30).
> >>- On ImageNet, it achieves comparable performance (69.52) to the second-best baseline (69.54).
> >>- Overall, while JCGEL (Soft) yields the best performance by allowing flexibility, JCGEL (Strict) still provides a strong improvement over conventional methods, confirming the robustness of our architectural core.
> >>\\begin{array}{|c|c|c|c|c|c|}
> \\hline
> & Pets & Flowers & Aircraft & STL10 & ImageNet \\\\
> \\hline
> 2^{nd} \text{ best} & 74.86(\pm 1.28) & 55.62(\pm 1.36) & 53.02(\pm 0.24) & 85.30(\pm 0.09) & 69.54 \\\\
> \text{JCGEL (strict)} & 75.25(\pm 0.61) & 54.61(\pm 0.10) & \textbf{54.61}(\pm 0.99) & 85.49(\pm 0.35) & 69.52 \\\\
> \text{JCGEL (soft)} & \textbf{77.51}(\pm 0.45) & \textbf{56.73}(\pm 1.37) & 54.11(\pm 0.83) & \textbf{85.54}(\pm 0.26) & \textbf{70.43} \\\\
> \\hline
> \\end{array}
>
> ---
>
> - **[Question 1]** Is there a difference between the subscripted  $\tau_g$ and $\tau_c$? The use seems to switch between the appendix and the main paper.
>
> > We apologize for the confusion caused by the notation inconsistency between the main text and the Appendix, and we have unified all references to $\tau_g$ in the revised manuscript.
> >
> > **Clarification**: In the previous Appendix C.4, we inadvertently used the notation $\tau_r$ to represent the temperature for rotation. We have corrected this to match the main text.
> >
> > **Standardized Notation**: In the revised paper, the parameters are consistently defined as:
> >- $\tau_c$: Temperature parameter controlling color bias.
> >- $\tau_g$: Temperature parameter controlling geometric (rotation) bias.

---

> ### Author Response · Authors · 2025-12-03
> **Response to Reviewer kNTr 4.**
>
> - **[Question 2]** In figure 3, are these two plots shown for the same training regime? If so, why is it the case that given JCGEL so clearly outperforms comparators on the cross entropy loss, that a greater difference is not observed in the evaluation accuracy. Is the testing set also long-tailed? Why does evaluation loss increase at all as training epoch increases? This is not clear to me.
>
> > **Response to Figure 3, Discrepancy between Loss and Accuracy** (Appendix C.4.2 with highlighted number 19 in revised paper): We confirm that 1) the plots in Figure 3 represent the same training regime, and 2) the test set follows a uniform (balanced) distribution, not a long-tailed one.
> >
> > The discrepancy—where JCGEL shows a significant drop in Cross Entropy (CE) loss compared to a moderate gain in accuracy—occurs because CE loss measures the quality of predicted probabilities (calibration), whereas Accuracy depends solely on the top-1 ranking.
> >- **Why Loss decreases more significantly than Accuracy**: A large drop in CE loss implies that our model assigns a significantly higher probability to the ground-truth class, even if this improvement does not immediately flip the top-1 ranking [1, 2]. To verify this, we decomposed the test loss into "Correctly Classified" and "Misclassified" subsets (see Table below).
> >>- **"Less Wrong" Predictions**: Notably, JCGEL (Soft) achieves a loss of 10.0870 on misclassified samples, which is drastically lower than Conv (17.9391) or E2CNN (18.5655).
> >>- This indicates that even when JCGEL fails to predict the top-1 class, it assigns a much higher probability to the true class compared to baselines (i.e., it is "less wrong"), resulting in a lower overall loss despite similar accuracy.
> >- **Why Evaluation Loss Increases**: The increase in evaluation loss at later epochs is due to overfitting to the head classes, a common phenomenon when training on long-tailed data and evaluating on a uniform test set [3, 4]. As the model minimizes training loss by becoming overconfident on frequent classes, it incurs a higher penalty (loss) on the balanced test set. However, JCGEL maintains a consistently lower loss than comparators, demonstrating superior robustness.
> > \\begin{array}{|c|c|c|c|c|c|c|c|}
> \\hline
> & \text{Conv.} & \text{E2CNN} & \text{CEConv} & \text{Hue-4-Sat-3} & \text{AE-Net} & \text{JCGEL (strict)} & \text{JCGEL (soft)} \\\\
> \\hline
> \text{right prediction} & 0.00492 & 0.0582 & 0.0319 & 0.0584 & 0.0759 & 0.0461 & 0.0491 \\\\
> \text{wrong prediction} & 17.9391 & 18.5655 & 19.0237 & 15.1587 & 33.3424 & 11.4870 & 10.0870 \\\\
> \\hline
> \\end{array}
> >
> > [1] Intra Order-Preserving Functions for Calibration of Multi-Class Neural Networks”, NeurIPS 2020
> >
> > [2] Soft Calibration Objectives for Neural Networks”, NeurIPS 2021.
> >
> > [3] Inducing Neural Collapse in Deep Long-tailed Learning Xuantong Liu et al., AISTATS 2023 (PMLR).
> >
> > [4] Long-Tailed Classification by Keeping the Good and Removing the Bad Momentum Causal Effect, Kaihua Tang et al., NeurIPS 2020.

---

### Official Review · Reviewer_mmgp · 2025-11-01

**Soundness:** 2
**Presentation:** 2
**Contribution:** 2
**Rating:** 2
**Confidence:** 5

**Summary:**

The authors introduce a neural network that is equivariant to hue transformations, and 90 degree rotations and reflections about the x and y axes (D_4). The network is designed according to the prescription offered in [1,2]. This problem is somewhat interesting, but as far as I understand, it isn’t particularly challenging. Experimental analysis validates the proposed solution.

[1] Cohen et al. ICML 2016
[2] Kondor et al. ICML 2018

**Strengths:**

**Originality.** The work appears original.

**Weaknesses:**

**Quality.** I rate the quality of this work as fair. The idea is somewhat interesting, but the mathematics are imprecise and should be reviewed, and the empirical analysis could be improved. Regarding the mathematics, some of the equations are quite difficult to parse. They should be made more reader friendly or paired with adequate written explanations. Regarding the experimental analysis, some experimental design decisions should be clarified (see questions/comments) and the baselines should include [2].

**Clarity.** The organization is fine, but the writing is not clear and should be reviewed (see notes in questions/comments and possible typos).

**Significance.** I rate the significance of this work as medium. While the utility of the work is clear from the experiments, the idea that matrix groups (whether for color or geometry) can be combined is unsurprising. This has been shown empirically in both in previous geometric equivariant works (e.g. [1]) and color equivariant works (e.g. [2]). Moreover, the product of two matrix groups is itself a matrix group and theoretical works have also shown convolutional neural networks can be designed for them (e.g. [3]).

[1] Esteves et al. ICLR 2018
[2] Yang et al. ICLR 2025
[3] Kondor et al. ICML 2018

**Questions:**

**Questions/Comments**
- Line 141: it probably makes sense to also define the group action \alpha as being an action on X here
- The notation in eq 3 is inconsistent. What is i?
- Line 153: this sentence is confusing. It would be helpful to separate the discussion about group convolution and the lifting layer.
- As far as I can tell this paper considers hue equivariance but not color equivariance. This should be clarified.
- The definition of the action of H is not provided. It is described as being a subgroup of SO(3) (line 183) but also defined as commutative (eq 8). The inconsistency is noted on Line 215 but not explained, please clarify.
- Line 259. I don’t understand what is being explained here.
- Line 290: What is the reason for varying n here?

**Possible typos**
- Line 033: works have been → works have
- Line 039:  including graph → including graphs
- Line 132: e x → \alpha(e, x)
- Line 151: and forming the group through integer summation → (delete)
- Line 159: refferred → referred
- Line 200: Here the hue shift is described as \mathbb{Z}^2
- Line 222: Then, composed of the above → The

---

> ### Author Response · Authors · 2025-12-03
> **Response to Reviewer mmgp 1.**
>
> - Dear Reviewer mmgp, Thank you for taking the time to review this manuscript and sharing thoughtful feedback. We added the corresponding content to the revised paper. The text is written in red, and the beginning is marked with a yellow-highlighted number.
>
> ---
>
>
> - **[Quality 1]** The idea is somewhat interesting, but the mathematics are imprecise and should be reviewed, and the empirical analysis could be improved.
> - **[Quality 2]** Regarding the mathematics, some of the equations are quite difficult to parse. They should be made more reader friendly or paired with adequate written explanations.
>
> > We respond both questions at **Question 1-5**.
>
> ---
>
> - **[Quality 3]** Regarding the experimental analysis, some experimental design decisions should be clarified (see questions/comments) and the baselines should include [2].
>
> > **Response to the suggestion on additional baselines**: We successfully integrate the requested baseline, Hue-4-Sat-3 [2], into our experimental suite, covering imbalanced environments, disentanglement learning and classification tasks.
> The updated quantitative results are presented in Tables 2–4 of the revised manuscript. As shown in the tables, our proposed model consistently outperforms the Hue-4-Sat-3 model across all evaluated settings. This comparison further validates the superiority of our approach in handling both color-geometry equivariance.
> >
> > [2] Yulong Yang, Felix O’Mahony, and Christine Allen-Blanchette. "Learning color equivariant representations." ICLR, 2025.
> > \\begin{array}{|c|c|c|c|c|c|c|}
> \\hline
> \text{3D Shapes} & \text{beta-VAE} & \text{FVM} & \text{MIG} & \text{SAP} & \text{DCI-Dis.} & \text{DCI-Com.} \\\\
> \\hline
> 2^{nd} \text{ best} & 82.67(\pm 3.06) & 83.88(\pm 1.44) & 44.74(\pm 8.16) & \textbf{9.15}(\pm 1.51) & 59.66(\pm 4.44) & 61.44(\pm 4.18) \\\\
> \text{Hue-3-Sat-4} & 80.00(\pm 9.17) & 79.88(\pm 2.25) & 28.59(\pm 6.89) & 5.90(\pm 1.51) & 45.94(\pm 4.61) & 47.66(\pm 4.97) \\\\
> \text{JCGEL} & \textbf{92.67}(\pm 7.02) & \textbf{87.67}(\pm 4.57) & \textbf{56.72}(\pm 3.94) & 8.55(\pm 1.90) & \textbf{66.86}(\pm 4.74) & \textbf{67.82}(\pm4.94) \\\\
> \\hline
> \\end{array}
> \\begin{array}{|c|c|c|c|c|c|c|}
> \\hline
> \text{MPI3D} & \text{beta-VAE} & \text{FVM} & \text{MIG} & \text{SAP} & \text{DCI-Dis.} & \text{DCI-Com.} \\\\
> \\hline
> 2^{nd} \text{ best} & 58.00(\pm 7.21) & 41.44(\pm 5.21) & 3.85(\pm 0.51) & 2.57(\pm 0.88) & 21.60(\pm 1.18) & 27.68(\pm 1.08) \\\\
> \text{Hue-3-Sat-4} & 51.33(\pm 1.15) & 42.42(\pm 6.25) & 5.05(\pm 1.52) & 3.37(\pm 0.74) & 20.77(\pm 0.86) & 28.65(\pm 0.90) \\\\
> \text{JCGEL} & \textbf{60.67}(\pm 2.31) & \textbf{45.75}(\pm 3.56) & \textbf{12.27}(\pm 2.05) & \textbf{6.20}(\pm 5.89) & \textbf{23.27}(\pm 4.04) & \textbf{31.93}(\pm 5.56) \\\\
> \\hline
> \\end{array}
> \\begin{array}{|c|c|c|c|c|c|}
> \\hline
> & Pets & Flowers & Aircraft & STL10 & ImageNet \\\\
> \\hline
> 2^{nd} \text{ best} & 74.86(\pm 1.28) & 55.62(\pm 1.36) & 53.02(\pm 0.24) & 85.30(\pm 0.09) & 69.54 \\\\
> \text{Hue-3-Sat-4} & 65.39(\pm 3.42) & 54.16(\pm 0.61) & 52.95(\pm 2.66) & 79.23(\pm 0.19) & 64.45 \\\\
> \text{JCGEL} & \textbf{77.51}(\pm 0.45) & \textbf{56.73}(\pm 1.37) & \textbf{54.11}(\pm 0.83) & \textbf{85.54}(\pm 0.26) & \textbf{70.43} \\\\
> \\hline
> \\end{array}
>
> ---
>
> - **Clarity.** The organization is fine, but the writing is not clear and should be reviewed (see notes in questions/comments and possible typos).
>
> > We respond at **Question**.

---

> ### Author Response · Authors · 2025-12-03
> **Response to Reviewer mmgp 2.**
>
> - **[Significance 1]** I rate the significance of this work as medium. While the utility of the work is clear from the experiments, the idea that matrix groups (whether for color or geometry) can be combined is unsurprising. This has been shown empirically in both in previous geometric equivariant works (e.g. [12]) and color equivariant works (e.g. [11]). Moreover, the product of two matrix groups is itself a matrix group and theoretical works have also shown convolutional neural networks can be designed for them (e.g. [13]).
> [12] Esteves et al. ICLR 2018 [11] Yang et al. ICLR 2025 [13] Kondor et al. ICML 2018.
> >
> > **(Response)** While we acknowledge that the combination of matrix groups is theoretically well-defined, our significant contribution lies in bridging the gap between abstract theory and practical application.
> We respectfully emphasize that theoretical possibility does not automatically translate to practical utility.
> >- Gap in Prior Works: As noted, previous works handled  geometric [1-9, 12] and color [11]  equivariance in isolation. Furthermore, while theoretical frameworks like [13] suggest the feasibility of product groups, they lack concrete implementation and empirical validation for this specific joint action.
> >- Our Contribution: To the best of our knowledge, our work is the first to successfully implement and empirically validate a model that is simultaneously equivariant to both color and geometric transformations. By demonstrating its utility across multiple computer vision tasks, we establish the practical significance of this theoretical concept.
> >
> > \\begin{array}{|c|c|c|}
> \\hline
> \text{Method} & \text{Color-Equivaraint} & \text{Geometric-Equivariant} \\\\
> \\hline
> [1-9, 12] & X & \checkmark \\\\
> [10, 11] & \checkmark & X \\\\
> \text{JCGEL (Orus)} & \checkmark & \checkmark \\\\
> \\hline
> \\end{array}
> >
> > We add this content to the Introduction (**Lines 73–75** with highlighted number 1 in revised paper).
> >
> > [1] Taco Cohen and M. Welling. Group equivariant convolutional networks. ICML, 2016.
> >
> > [2]  Daniel E. Worrall, Stephan J. Garbin, Daniyar Turmukhambetov, and G. Brostow. Harmonic networks: Deep translation and rotation equivariance. CVPR, 2016.
> >
> > [3] Taco S. Cohen and Max Welling. Steerable cnns. CoRR, abs/1612.08498, 2016.
> >
> > [4] Maurice Weiler, F. Hamprecht, and M. Storath. Learning steerable filters for rotation equivariant cnns. CVPR, 2017.
> >
> > [5] Ivan Sosnovik, Michal Szmaja, and A. Smeulders. Scale-equivariant steerable networks. ICLR, 2019.
> >
> > [6] Daniel E. Worrall and Max Welling. Deep scale-spaces: Equivariance over scale. NeurIPS 2019.
> >
> > [7] Maurice Weiler and Gabriele Cesa. General e(2)-equivariant steerable cnns. NeurIPS 2019.
> >
> > [8] L. MacDonald, Sameera Ramasinghe, and S. Lucey. Enabling equivariance for arbitrary lie groups. CVPR 2021.
> >
> > [9] Weizheng Qiao, Yang Xu, and Hui Li. Scale-rotation-equivariant lie group convolution neural networks (lie group-cnns). In arXiv.org, 2023.
> >
> > [10] A. Lengyel, Ombretta Strafforello, Robert-Jan Bruintjes, Alexander Gielisse, and Jan van Gemert. Color equivariant convolutional networks. NeurIPS 2023.
> >
> > [11] Yulong Yang, Felix O’Mahony, and Christine Allen-Blanchette. Learning color equivariant repre-
> sentations. ICLR 2025.
> >
> > [12] Esteves C, Allen-Blanchette C, Zhou X, Daniilidis K. Polar Transformer Networks. International Conference on Learning Representations. 2018.
> >
> > [13] Kondor R, Trivedi S. On the Generalization of Equivariance and Convolution in Neural Networks to the Action of Compact Groups. International Conference on Machine Learning. 2018.

---

> ### Author Response · Authors · 2025-12-03
> **Response to Reviewer mmgp 3.**
>
> - **[Questions / Comments 1]** Line 141($\to$ 146 in revised paper): it probably makes sense to also define the group action \alpha as being an action on X here
>
> > **(Response)** Yes, that is true we add group action \alpha in  ‘Equivariant Map.’ paragraph in the revised paper (Line 146 with highlighted number 2).
>
> ---
>
> - **[Questions / Comments 2]** The notation in eq 3 is inconsistent. What is i?
>
> > **(Response)** The notation $\psi^{l, i}$ denotes that is an i^{th} kernel of l^{th} layer and we add this notation right under the Eq. 3 in the revised version (Line 166 with highlighted number 5).
>
> ---
>
> - **[Questions / Comments 3]** Line 153($\to$ 161-182 in revised paper): this sentence is confusing. It would be helpful to separate the discussion about group convolution and the lifting layer.
>
> > **(Response)**  We restructure the section to clearly separate the discussion of ‘Group Convolution’ and ‘Lifting Layer’ as suggested (This revision is in Lines 161-182 with highlighted numbers 5-8 in revised paper.). Specifically, we reorganized the text into three distinct parts: Standard Convolution, Lifting Layer, and Group Convolution. In the revised paper, we clarified that the term $y-x$ represents a discrete translation shift in standard convolution (addressing the confusion in the original Line 153) and explicitly described how this concept generalizes to other groups in the Lifting and Group Layers.
>
> ---
>
> - **[Questions / Comments 4]** As far as I can tell this paper considers hue equivariance but not color equivariance. This should be clarified.
>
> > **(Response)** Yes, we consider the hue-shift equivaraince rather including saturation.
>
> ---
>
> - **[Questions / Comments 5]** The definition of the action of H is not provided. It is described as being a subgroup of SO(3) (line 183 $\to$ 196, 227 in revised paper) but also defined as commutative (eq 8). The inconsistency is noted on Line 215 but not explained, please clarify.
>
> > **(Response)** We clarify the definitions and adopted distinct notations for the group actions in the lifting and group layers to resolve the ambiguity. The perceived inconsistency arose from using the same symbol for different actions. We revise the manuscript as follows (Lines 196 (Lifting layer) and 227 (Group layer) with highlighted number 9 in revised paper):
> >- **Lifting layer ($H_n(m)$)**: we clarify that this action acts as a subgroup of $SO(3)$, performing a channel-wise rotation (hue-shift) on the RGB input.
> >- **Group layer ($\mathbb{H}_n(m)$)**: We introduce a notation $\mathbb{H}_n(m)$ to explicitly define the action in the group convolution layer. This is defined as a cyclic group action (modulo $n$).
>
> ---
>
> - **[Questions / Comments 6]** Line 259($\to$ 270-275 in revised paper). I don’t understand what is being explained here.
>
> > **(Response)** We revise the paragraph to cleary explain that it describes the formulation of the soft-equivariant version of JCGEL (Line 270-275 with highlighted number 10 in revised paper). Specifically, we clarified that this approach allows the model to handle environments with imperfect symmetries, which improves generalization as discussed in the 'Related Work' section. We explicitly stated that we adapted the re-weighting mechanism from [1] to implement soft-equivariant JCGEL, enabling a direct performance comparison between strict- and soft-equivariant JCGEL.
> >
> > [1] David W. Romero and M. Hoogendoorn. Co-attentive equivariant neural networks: Focusing equivariance
> on transformations co-occurring in data. In International Conference on Learning Representations,
> 2019.
>
> ---
>
> - **[Questions / Comments 7]** Line 290 ($\to$ 307-309 in revised paper) : What is the reason for varying n here?
>
> > **(Response)** We vary $n$ (spatial resolution) to verify the robustness of our method across different feature map sizes, which is crucial for practical CNN architectures (revised manuscript in Lines 307-309 with highlighted number 11). Since CNNs hierarchically process feature maps at varying scales (due to pooling or striding), it is essential to demonstrate that our proposed operation remains effective regardless of the spatial dimensions.

---

> ### Author Response · Authors · 2025-12-03
> **Response to Reviewer mmgp 4.**
>
> - **Possible typos**
> >- Line 033: works have been → works have
> >- Line 039: including graph → including graphs
> >- Line 132: e x → \alpha(e, x)
> >- Line 151: and forming the group through integer summation → (delete)
> >- Line 159: refferred → referred
> >- Line 200: Here the hue shift is described as \mathbb{Z}^2
> >- Line 222: Then, composed of the above → The
> >>- **(Response)** We fix all the typos and notation errors pointed out. Regarding Line 200, we have corrected the notation for the hue shift to $\mathbb{Z}_n$ (Cyclic Group ) to accurately reflect its definition. As defined in CEConv [1], the hue shift corresponds to a rotation in the color space governed by the matrix $H_n(m)$, which warrants the $\mathbb{Z}_n$ notation:
> $$ H_n(m) = \begin{bmatrix} \cos(\frac{2m\pi}{n}) + a & a - b & a + b \\\\ a + b & \cos(\frac{2m\pi}{n}) + a & a - b \\\\ a - b & a + b & \cos(\frac{2m\pi}{n}) + a \end{bmatrix} $$

---

### Official Review · Reviewer_K3eD · 2025-11-03

**Soundness:** 2
**Presentation:** 2
**Contribution:** 2
**Rating:** 2
**Confidence:** 5

**Summary:**

This paper proposes a “joint color and geometric equivariant convolutional layer” by combining a standard D4 geometric group with a cyclic color-transformation group, which simply forms a direct-product symmetry. The construction reduces to weight-sharing across these commuting actions, yielding what is essentially a simple combination of existing geometric group-equivariant layers and color-equivariant layers. Experiments are conducted on synthetic equivariance tests, imbalanced MNIST, disentanglement benchmarks, and several image-classification datasets

**Strengths:**

1. The paper is easy to follow.
2. Experiments are conducted to demonstrate that intended equivariance is indeed achieved on synthetic dataset

**Weaknesses:**

1. Since the color transformation group and the $D_4$ geometric group action commute, the method effectively reduces to a direct-product construction. As a result, the paper simply combines (Cohen & Welling, 2016) and (Lengyel et al., 2023) in a straightforward and largely trivial manner: lift the input to a larger group and apply group convolution in the lifted space. There is no conceptual novelty.
2. All technical components used in the paper are standard, and the work does not provide any meaningful theoretical or architectural contribution.
3. The experimental section is also weak. Demonstrating equivariance on synthetic data only verifies that the implementation behaves as expected. The claimed gains on real datasets are unconvincing: (i) there is no discussion of the computational overhead caused by performing group convolution on a larger product group, and (ii) the empirical improvements are marginal (e.g., on CIFAR-100). In fact, publicly reported results under similar training setups substantially exceed those shown here; for example, see the benchmarks: https://openmixup.readthedocs.io/en/latest/mixup_benchmarks/Mixup_cifar.html?utm_source=chatgpt.com

**Questions:**

Please see the previous section

---

> ### Author Response · Authors · 2025-12-03
> **Response to Reviewer K3eD 1.**
>
> - Dear Reviewer K3eD, Thank you for taking the time to review this manuscript and sharing thoughtful feedback. We added the corresponding content to the revised paper. The text is written in red, and the beginning is marked with a yellow-highlighted number.
>
> ---
>
> - **[Weakness 1]** Since the color transformation group and the geometric group action commute, the method effectively reduces to a direct-product construction. As a result, the paper simply combines (Cohen & Welling, 2016) and (Lengyel et al., 2023) in a straightforward and largely trivial manner: lift the input to a larger group and apply group convolution in the lifted space. There is no conceptual novelty.
> - **[Weakness 2]** All technical components used in the paper are standard, and the work does not provide any meaningful theoretical or architectural contribution.
>
> > **Response to the concern regarding novelty and triviality**: We clarify that our primary contribution lies in the **first successful architectural realization and comprehensive validation** of the joint color-geometric equivariance, **bridging the gap between theoretical possibility and practical utility.** While the mathematical concept of a direct product is established, translating it into a working deep learning model involves non-trivial challenges that prior works have not addressed. Furthermore, this gap is critically large in practice, remaining unreported in the literature, even though it may look conceptually simple to researchers with reviewers’ background.
> >
> > Here is a more clear contribution summary.
> >
> > **[Architectural Novelty (Solving the Implementation Gap)]**
> >- **Challenge**: As the reviewer noted, the theoretical group is a direct product $(\mathbb{Z}^2 \rtimes G_o) \times G_c$. However, simply combining existing convolutions often leads to channel interference where color and geometric features become entangled, degrading the equivariant property.
> >- **Our Solution**: We designed a specific architectural mechanism to ensure that color and geometric operations remain computationally independent across channels. This implementation is not a trivial combination but a careful engineering of the group interaction to maintain strict equivariance, which has not been realized in previous literature despite theoretical discussions, as shown in the Table.
> >
> > \\begin{array}{|c|c|c|}
> \\hline
> \text{Method} & \text{Color-Equivaraint} & \text{Geometric-Equivariant} \\\\
> \\hline
> [1-9] & X & \checkmark \\\\
> [10, 11] & \checkmark & X \\\\
> \text{JCGEL (Orus)} & \checkmark & \checkmark \\\\
> \\hline
> \\end{array}
> >
> > We add this content to the Introduction (**Lines 73–75** with highlighted number 1 in revised paper).
> >
> > [1] Taco Cohen and M. Welling. Group equivariant convolutional networks. ICML, 2016.
> >
> > [2]  Daniel E. Worrall, Stephan J. Garbin, Daniyar Turmukhambetov, and G. Brostow. Harmonic networks: Deep translation and rotation equivariance. CVPR, 2016.
> >
> > [3] Taco S. Cohen and Max Welling. Steerable cnns. CoRR, abs/1612.08498, 2016.
> >
> > [4] Maurice Weiler, F. Hamprecht, and M. Storath. Learning steerable filters for rotation equivariant cnns. CVPR, 2017.
> >
> > [5] Ivan Sosnovik, Michal Szmaja, and A. Smeulders. Scale-equivariant steerable networks. ICLR, 2019.
> >
> > [6] Daniel E. Worrall and Max Welling. Deep scale-spaces: Equivariance over scale. NeurIPS 2019.
> >
> > [7] Maurice Weiler and Gabriele Cesa. General e(2)-equivariant steerable cnns. NeurIPS 2019.
> >
> > [8] L. MacDonald, Sameera Ramasinghe, and S. Lucey. Enabling equivariance for arbitrary lie groups. CVPR 2021.
> >
> > [9] Weizheng Qiao, Yang Xu, and Hui Li. Scale-rotation-equivariant lie group convolution neural networks (lie group-cnns). In arXiv.org, 2023.
> >
> > [10] A. Lengyel, Ombretta Strafforello, Robert-Jan Bruintjes, Alexander Gielisse, and Jan van Gemert. Color equivariant convolutional networks. NeurIPS 2023.
> >
> > [11] Yulong Yang, Felix O’Mahony, and Christine Allen-Blanchette. Learning color equivariant repre-
> sentations. ICLR 2025.

---

> ### Author Response · Authors · 2025-12-03
> **Response to Reviewer K3eD 2.**
>
> - **[Weakness 3]** The experimental section is also weak. 1) Demonstrating equivariance on synthetic data only verifies that the implementation behaves as expected. The claimed gains on real datasets are unconvincing: 2) there is no discussion of the computational overhead caused by performing group convolution on a larger product group.
>
> > **Response to the concerns regarding experimental strength and computational cost**:
> We have strengthened the experimental section along three axes: **1)** clarifying the role of synthetic tests and adding challenging real-world validations, **2)** providing a detailed analysis of computational overhead, and **3)** demonstrating that the empirical gains are statistically significant under a controlled training protocol. We also compare to the dataset augmentation method and clarify why our absolute CIFAR-100 numbers are not directly comparable to mixup-optimized benchmarks such as OpenMixup.
> >
> >1. **Role of synthetic experiments** (Table 2) **and additional real-data validation** (We add this content to the Appendix C.4.1 Appendix C.4.1 Lines 1079 highlighted number 18 & Table 6 in the revised paper). We agree that synthetic equivariance tests primarily verify that the implementation behaves as expected. Our intent with these experiments is precisely to isolate the group action and confirm that each channel responds independently to color and geometric transformations. To address your concern, we now explicitly position these experiments as sanity checks and complement them with more realistic evaluations.
> >>- **Joint Bias Stress-Test**: We introduce a biased color-rotated MNIST setting where both hue and rotation are simultaneously biased. Unlike prior works that typically probe robustness under either color-biased or rotation-biased conditions separately, this new setting stresses the joint color–geometry equivariance of our layer.
> >>- **Real-World Expansion**: We expand our evaluations to long-tailed and imbalanced datasets, including ImageNet-LT and fine-grained benchmarks. In these tasks, our jointly equivariant model consistently outperforms geometric-only and color-only baselines under the same backbone and training regime (see revised Tables 4-6), indicating that the benefits extend beyond synthetic data.
> >>
> >> \\begin{array}{|c|c|c|c|c|c|}
> \\hline
> & CNN & E2CNN & CEConv &AE-Net & JCGEL (Ours) \\\\
> \\hline
> \text{ImageNet-LT} & 30.93 & 31.25 & 31.19 & 29.22 & \textbf{31.65} \\\\
> \\hline
> \\end{array}
> >
>
> >2. **Computational overhead of the product-group convolution.** We have added a dedicated subsection reporting the computational cost, specifically measuring training/inference throughput and Peak GPU Memory (VRAM) consumption, normalized to the standard CNN baseline ($\times 1.00$). As summarized in the revised manuscript, our JCGEL layer results in lower training throughput ($\approx \times 0.14$) and higher memory consumption ($\approx \times 4.37$). We openly acknowledge that operating on the larger product group $(\mathbb{Z}^2 \rtimes G_o) \times H_n$ inevitably incurs these computational costs. However, we demonstrate that this additional cost translates directly into superior performance and robustness (as shown in Tables 2-5), which standard efficient models cannot achieve. We have explicitly documented this trade-off as a current limitation and a priority for future optimization in the "Limitations" section (Appendix D with highlighted number 21  in the revised paper).
> >
> > \\begin{array}{|c|c|c|c|c|c|}
> \\hline
> & \text{CNN} & \text{CEConv} & \text{E2CNN} & \text{Hue-4-Sat-3} & \text{JCGEL (Ours)} \\\\
> \\hline
> \text{Training Speed} & \times 1.00 & \times 0.84 & \times 0.31 & \times 0.12 & \times 0.14 \\\\
> \text{Inference Speed} & \times 1.00 & \times 0.72 & \times 0.27 & \times 0.09 & \times 0.09 \\\\
> \text{VRAM Usage} & \times 1.00 & \times 1.37 & \times 1.74 & \times 2.04 & \times 4.37 \\\\
> \\hline
> \\end{array}

---

> ### Author Response · Authors · 2025-12-03
> **Response to Reviewer K3eD 3.**
>
> - **[Weakness 3]**  **3)** the empirical improvements are marginal (e.g., on CIFAR-100). In fact, publicly reported results under similar training setups substantially exceed those shown here; for example, see the benchmarks: https://openmixup.readthedocs.io/en/latest/mixup_benchmarks/Mixup_cifar.html?utm_source=chatgpt.com
>
> > **Response regarding the comparison with OpenMixup benchmarks and the magnitude of improvements**.
> To address the concern that our gains might be marginal, we performed paired statistical tests between our method and the second-best baseline for each classification dataset (Appendix C.6.2 and Table 9 with highlighted number 20 in the revised paper). The resulting p-values are: EuroSAT: $0.979$, CIFAR-100: $0.029$, Pets: $0.008$, Flowers: $0.019$, Aircraft: $0.003$, STL-10: $0.011$, Food-101: $0.001$.
> Except for EuroSAT, all datasets yield $p < 0.05$, indicating that the improvements are statistically significant rather than due to randomness. Moreover, the same pattern holds for our imbalanced and disentanglement experiments, where our model consistently outperforms the strongest baselines.
> >
> > \\begin{array}{|c|c|c|c|c|c|c|c|}
> \\hline
> & \text{EuroSAT} & \text{CIFAR100} & \text{Pets} & \text{Flowers} & \text{Aircraft} & \text{STL10} & \text{Food101} \\\\
> \\hline
> 2^{nd} \text{ best} & 97.83(\pm 0.15) & 77.29(\pm 0.01) & 74.86(\pm 1.28) & 55.62(\pm 1.36) & 53.02(\pm 0.24) & 85.30(\pm 0.09) & 81.45(\pm 0.31) \\\\
> \text{JCGEL} & 97.70(\pm 0.18) & 77.51(\pm 0.45) & 76.08(\pm 0.80) & 56.73(\pm 1.37) & 54.11(\pm 0.92) & 85.54(\pm 0.26) & 82.62(\pm 0.38) \\\\
> \text{p-value} & 0.979 & \textbf{0.029} & \textbf{0.008} & \textbf{0.019} & \textbf{0.003} & \textbf{0.011} & \textbf{0.001} \\\\
> \\hline
> \\end{array}
> >
> > **Regarding the comparison with OpenMixup benchmarks**: While benchmarks like OpenMixup achieve high absolute numbers using heavy data augmentation (Mixup, CutMix), our controlled experiments show that JCGEL outperforms baselines even when they are trained with standard augmentations.
> >
> >-  **Comparison with Data Augmentation (ImageNet Results)**: To address the concern that our gains might be marginal compared to augmentation strategies, we evaluated a standard ConvNet trained with the ImageNet augmentation policy (Conv.+Aug) and we add this content in Table 4 in the revised paper.
> >>- As shown in the table below, JCGEL (Soft, 70.43%) significantly outperforms Conv.+Aug (66.64%).
> >>- This result is crucial: it proves that intrinsic architectural equivariance provides a more effective inductive bias than trying to learn invariance solely through data augmentation.
> >> \\begin{array}{|c|c|c|c|c|c|c|c|}
> \\hline
> & \text{Conv.} & \text{Conv.+Aug.} & \text{CEConv} & \text{E2CNN} & \text{JCGEL (strict)} & \text{AE-Net} & \text{JCGEL (soft)} \\\\
> \\hline
> \text{ImageNet} & 64.77 & 66.64 & 67.83 & 64.73 & 69.52 & 69.54 & \textbf{70.43} \\\\
> \\hline
> \\end{array}
>
> >- **Distinction from Engineering Benchmarks**: We note that benchmarks like OpenMixup are designed to achieve state-of-the-art accuracy using strong augmentations and carefully tuned recipes. In contrast, our experiments deliberately use a unified, simple training protocol for all baselines to isolate the intrinsic effect of color–geometry equivariance. We have clarified this distinction in the experimental section, emphasizing that our goal is a controlled architectural comparison rather than competing with highly optimized engineering pipelines on absolute accuracy.

---

### Note · Authors · 2026-02-04

I have read and agree with the venue's withdrawal policy on behalf of myself and my co-authors.

---

### Meta-Review · Area_Chair_7snL · 2026-01-06

**Summary:**

The paper introduces JCGEL, a joint color‑and‑geometry equivariant convolution layer achieved through weight sharing across commuting group actions. Multiple reviewers express concerns about the significance of combining geometric and color equivariance, though the AC acknowledges the authors’ clear contribution in the architectural realization and comprehensive validation of joint color‑geometric equivariance.

**Reviewer Concerns:**

One major concern shared by multiple reviewers is that the color transformation group and the geometric group can be combined through a direct‑product construction. Studies have been demonstrated empirically in both prior geometric‑equivariant and color‑equivariant works. During the rebuttal, the authors clarified that their primary contribution lies in the architectural realization and comprehensive validation of joint color‑geometric equivariance. One reviewer also noted that some of the mathematical formulations are imprecise and should be reviewed, and that the empirical analysis could be strengthened. The AC appreciates the authors’ efforts during rebuttal in carefully responding to these comments and making corresponding revisions.

**Reviewer Scores:**

This paper receives the following ratings: Reject, Reject, Marginally Above, and Marginally Below. If the reviewers had been able to participate fully in the discussion, the AC would expect some negative ratings to remain. The AC recommends not accepting the paper.

---

### Decision · Program_Chairs · 2026-01-26

Reject